# Prickle and Ror modulate Dishevelled-Vangl interaction to regulate non-canonical Wnt signaling during convergent extension in *Xenopus*

Hwa-seon Seo[1†‡], Deli Yu[1†], Ivan K Popov[1], Jiahui Tao[1], Allyson R Angermeier[1], Fei Yang[1§], Sylvie Marchetto[2], Jean-Paul Borg[2,3], Bingdong Sha[1], Jeffrey D Axelrod[4], Chenbei Chang[1], Jianbo Wang[1]*

[1]Department of Cell, Developmental and Integrative Biology, University of Alabama at Birmingham, Birmingham, United States; [2]Aix Marseille Univ, CNRS, INSERM, Institut Paoli-Calmettes, CRCM, Equipe labellisée Ligue 'Cell Polarity, Cell Signaling And Cancer', Marseille, France; [3]Institut Universitaire de France, Paris, France; [4]Department of Pathology, Stanford University School of Medicine, Stanford, United States

*For correspondence:
j18wang@uab.edu

†These authors contributed equally to this work

Present address: ‡iCura Diagnostics, Malvern, United States; §Chronic Disease Research Institute, School of Public Health, School of Medicine, Zhejiang University, Hangzhou, China

## eLife Assessment

This **valuable** study addresses mechanisms of feedback inhibition between planar cell polarity protein complexes during convergent extension movements in Xenopus embryos. The authors propose a conceptually new model, in which non-canonical Wnt ligand stimulates the transition of Dishevelled from its complex with Vangl to Frizzled, with essential roles of Prickle and Ror in this process. The main observations supporting molecular interactions rely on modest but significant changes in protein association in response to Wnt11. While the study is limited due to insufficient phenotypic analysis at the cellular level and the use of exogenously supplied proteins, this work is **convincing** and will be of broad interest to cell and developmental biologists.

**Abstract** Convergent extension (CE) is a fundamental morphogenetic process where oriented cell behaviors lead to polarized extension of diverse tissues. In vertebrates, regulation of CE requires both non-canonical Wnt, its co-receptor Ror, and several 'core members' of the planar cell polarity (PCP) pathway. PCP was originally identified as a mechanism to coordinate the cellular polarity in the plane of static epithelium, where core proteins Frizzled (Fz)/Dishevelled (Dvl) and Van Gogh-like (Vangl)/Prickle (Pk) partition to opposing cell cortex. But how core PCP proteins interact with each other to mediate non-canonical Wnt/Ror signaling during CE is not clear. We found previously that during CE, Vangl cell-autonomously recruits Dvl to the plasma membrane and keeps Dvl inactive. In this study, we show that non-canonical Wnt induces Dvl to transition from Vangl to Fz in *Xenopus* embryos. Pk inhibits the transition and functionally synergizes with Vangl to suppress Dvl during CE. Conversely, Ror is required for the transition and functionally antagonizes Vangl. Biochemically, Vangl interacts directly with both Ror and Dvl. Ror and Dvl do not bind directly but can be co-fractionated with Vangl. Collectively, we propose that Pk assists Vangl to function as an unconventional adaptor that brings Dvl and Ror into a complex to serve two functions: (1) simultaneously preventing both Dvl and Ror from ectopically activating non-canonical Wnt signaling; and (2) relaying Dvl to Fz for signaling activation upon non-canonical Wnt-induced dimerization of Fz and Ror.

## Introduction

Throughout the animal kingdom, convergent extension (CE) is a universal morphogenetic engine that reshapes tissues during embryogenesis (*Davey and Moens, 2017*; *Goodrich and Strutt, 2011*; *Huebner and Wallingford, 2018*; *Keller, 2002*). Through polarized cell intercalation, directional cell migration, or oriented cell division, CE generates powerful morphogenetic force to elongate a tissue in one direction while simultaneously narrowing it in the perpendicular direction. Disruption of CE can disturb normal embryogenesis from flies to mammals and cause various congenital disorders, including neural tube defects and skeletal disorders, such as Robinow Syndrome and Brachydactyly type B (*Butler and Wallingford, 2017*; *Wang et al., 2012*; *Yang and Mlodzik, 2015*).

In vertebrates, CE is regulated by genes in the planar cell polarity (PCP) pathway. PCP refers to cell polarity orthogonal to that of apical-basal in epithelial cells. It was initially discovered in *Drosophila* as a signaling mechanism coordinating polarized cellular structures in the plane of the epithelium. The PCP pathway consists of six 'core' proteins in flies, including three transmembrane proteins (the atypical cadherin Flamingo (Fmi), the receptor Frizzled (Fz), and the four-pass transmembrane protein Van Gogh (Vang; Vangl in vertebrates)), and three cytoplasmic proteins (Dishevelled (Dsh; Dvl in mammals), Diego (Dgo), and Prickle (Pk)). A key feature of the core PCP proteins is that they assemble into two distinct complexes, those of Fmi/Fz/Dsh/Dgo and Fmi/Vang/Pk, that localize asymmetrically on opposing cell cortexes. Extensive genetic and imaging studies in flies, combined with computational modeling, have led to a model of feedback interaction in establishing core PCP protein distribution. The model proposes that Fmi on neighboring cells can establish homophilic interaction to facilitate cross talk between extracellular Fz and Vang in trans at the cell-cell junctions, with Dsh, Dgo, and Pk functioning to stabilize the interacting complexes across the cell junctions. At the same time, these cytoplasmic proteins destabilize the juxtaposition of Fz and Vang in the same cell to segregate the complexes. Several mechanisms are used to facilitate both positive and negative feedback regulations, including selective interaction with partner proteins, post-translational modification of different components, stability of core proteins at cell junctions, transport of components along cytoskeletons, and membrane protein recycling and subcellular localization (*Cho et al., 2015*; *Ressurreição et al., 2018*; *Shimada et al., 2006*; *Strutt et al., 2013*; *Warrington et al., 2017*). These feedback mechanisms act together to promote asymmetric clustering of Fz/Dsh/Dgo and Vang/Pk complexes at the distal and proximal cell junctions, respectively, to regulate asymmetric cytoskeletal organization and to coordinate planar polarity across the entire epithelium (*Amonlirdviman et al., 2005*; *Axelrod and Tomlin, 2011*; *Humphries and Mlodzik, 2018*; *Strutt et al., 2016*).

Though the PCP components are conserved in vertebrates, they coordinate cellular polarity not only in the plane of epithelial cells, but also in actively migrating cells, including neurons, neural crest, metastatic cancer cells, and cells undergoing CE. These cells share dynamic behaviors with constantly changing cell-cell contacts and interactions. Asymmetric localization of individual PCP proteins has been reported in a number of such cells, but the pattern varies and segregation of PCP complexes has not been consistently observed (reviewed in *Davey and Moens, 2017*). Moreover, as the majority of these studies focus on the activities of individual PCP components in regulating polarized cell behaviors, less is known about how PCP proteins interact with each other to coordinately control the migratory processes (*Davey and Moens, 2017*).

Potential differences in PCP protein interaction and function in *Drosophila* and vertebrates have emerged from some recent studies. Vangl1/2 were reported to function in a Celsr-independent manner to regulate mammalian airway morphogenesis (*Paramore et al., 2024*). On the other hand, whereas mutual inhibition between *Vangl* and *Fz/Dvl* is expected based on the *Drosophila* work, a number of reports also reveal a functional synergy between these proteins in mice. For instance, a simultaneous decrease in gene dosage of *Vangl2* and *Dvl* enhanced CE defects in neural tube closure and cochlea elongation, and compound mouse mutants in *Vangl2* and several *Fz* genes show similar, more exacerbated defects than those with mutations in individual genes (*Etheridge et al., 2008*; *Wang et al., 2006*; *Yu et al., 2010*; *Yu et al., 2012*). Upstream of Fz, the requirement of Wnt ligand in fly PCP signaling was debated initially (*Chen et al., 2008*; *Wu et al., 2013*) and disproved more recently (*Ewen-Campen et al., 2020*; *Yu et al., 2020*). In contrast, non-canonical Wnts, including Wnt5a and 11, are essential for CE in vertebrates (*Grumolato et al., 2010*; *Heisenberg et al., 2000*; *Yamaguchi et al., 1999*), and a functional synergy between Vangl2 and Wnt5a has been shown in the development of multiple tissues in mice (*Gao et al., 2011*; *Qian et al., 2007*; *Sinha et al., 2012*;

*Wang et al., 2011*). These findings, which collectively imply a positive role of Vangl in Wnt/Fz/Dvl-mediated non-canonical Wnt signaling activation, bring up an essential question on how Vangl may both cooperate with and inhibit non-canonical Wnt/Fz/Dvl.

A further complication of non-canonical Wnt/PCP signaling during vertebrate development is the involvement of several co-receptors, including Ror1/2, Ptk7, and Ryk, whose functions are not linked to fly PCP (*Ripp et al., 2018*; reviewed in *Green et al., 2014*). For instance, Ror2 has been shown to bind to Wnt5a together with Fz and is required to mediate Wnt5a-induced phosphorylation of Dvl in mammals (*Grumolato et al., 2010*; *Ho et al., 2012*; *Nishita et al., 2010*). Mouse mutants deficient in Ror1/2 phenocopy many defects of *Wnt5a* mutants (*Ho et al., 2012*), demonstrating a critical function of this co-receptor family in Wnt5a/PCP signaling. Intriguingly, reduced gene dosages of both *Ror2* and *Vangl2* can lead to more severe morphogenesis defects than mutants of each individual gene, revealing a functional synergy between Vangl2 and Ror2 (*Gao et al., 2011*). This is reminiscent of the functional synergy observed between Vangl2 and non-canonical Wnt/Fz/Dvl. Ror2 was reported to interact with Vangl2 biochemically and proposed to form a receptor complex with Vangl2 in response to Wnt5a (*Gao et al., 2011*). However, the biochemical and cell biological activities of the Ror2 /Vangl complex and how this may affect Wnt/Fz/Dvl PCP signaling is not understood in detail.

To understand feedback regulation of core PCP proteins in vertebrate CE, we have used the mouse and the *Xenopus* models to investigate functional and biochemical interactions of these proteins. Our previous work suggested that Vangl2 has dual activity in modulating Dvl function: it binds and recruits Dvl to the plasma membrane cell-autonomously and keeps it inactive, but at the same time enriches Dvl at this subcellular domain for Fz signaling upon stimulation by the Wnt11 ligand, which triggers release of Dvl from Vangl2 (*Seo et al., 2017*). In the current study, we attempted to address two questions raised by this model: (1) how will Vangl's molecular partner Pk modulate Vangl-Dvl interaction during CE; and (2) if Dvl is sequestered by Vangl, how can it gain access to Fz in the response to Wnt?

In flies, Pk clusters with Vang to the proximal cell junction and is required to generate feedback amplification for asymmetric localization of both Vang and Dsh/Fz. Pk is shown to stabilize Fz-clusters on the plasma membrane in neighboring cells, but destabilizes Fz-clusters in a Dsh-dependent manner through endocytosis in the same cell (*Warrington et al., 2017*). Competitive binding of Pk to Dsh to prevent its plasma membrane recruitment by Fz has been proposed as a mechanism to destabilize Fz/Dsh clustering in the same cell (*Tree et al., 2002*). But binding between Pk and Dsh was reported to be quite weak (*Bastock et al., 2003*), and over-expressing Pk in *Xenopus* cannot effectively abolish Fz7-mediated recruitment of Dvl to the plasma membrane (*Takeuchi et al., 2003*; *Veeman et al., 2003*). These studies thus do not fully support the notion of competitive binding to Dvl as the underlying mechanism for Pk to destabilize Fz clusters or its action during CE.

In this study, we used gastrulating *Xenopus* embryos as a CE model to carry out functional, biochemical, and imaging studies and found that Pk helps Vangl2 to sequester both Dvl2 and Ror2, whereas Ror2 is needed for Dvl to transition from Vangl to Fz in response to non-canonical Wnt. We propose a novel model in which Pk assists Vangl to function as an unconventional adaptor that brings Dvl and Ror2 into a complex to serve two functions: (1) simultaneously preventing both Dvl and Ror2 from ectopic activation; and (2) relaying Dvl to Fz via Ror2 upon non-canonical Wnt activation. We propose that these two actions together help to modulate the threshold and dynamics of signaling activation in response to non-canonical Wnt.

## Results

### Pk synergizes with Vangl2 to suppress Dvl during CE

To probe the functional network of core PCP proteins during CE, we first studied how Pk interacts with Vangl2 and Dvl to regulate *Xenopus* body axis elongation in both gain- and loss-of-function scenarios. To overexpress Pk in *Xenopus* embryos, we used two different mRNAs that encode either a GFP-tagged mouse Pk2 (mPk2) or a Flag-tagged *Xenopus* Pk1 (XPk) (*Takeuchi et al., 2003*; *Vladar et al., 2012*). Similar to the previous report (*Takeuchi et al., 2003*), we found that injecting *Xpk* or *mPk2* mRNA into the dorsal marginal zone (DMZ) of 4 cell stage *Xenopus* embryos can block CE in a dose-dependent manner (*Figure 1—figure supplement 1a*). Quantification of the length-to-width ratio (LWR) indicates that 1 and 2 ng *pk* mRNA injection can reproducibly cause moderate and severe

CE defects, respectively, but 0.5 ng *Xpk* only results in a slight LWR reduction that is not statistically significant (*Figure 1—figure supplement 1a′*).

With co-injection of mRNAs, however, even a very small dose of 0.1 ng *Xpk* is sufficient to cause severe CE defects together with 0.1 ng mouse *Vangl2* (*mVangl2*), which produces only a moderate CE defect when injected by itself (*Figure 1a and a′*). Similarly, co-injecting 0.25 ng *mPk2*, which causes no CE defects by itself, also significantly enhances the CE defect induced by 0.1 ng *mVangl2* (*Figure 1b and b′*).

Conversely, we found that knocking down endogenous XPk level using antisense morpholino (*Xpk*MO, *Figure 1—figure supplement 1b–d′*) can rescue *mVangl2* over-expression induced CE defect (*Figure 1c and c′*), whereas over-expressing Pk can dose-dependently rescue *Xvangl2* morpholino (*XV*MO) knockdown-induced CE defect (*Figure 1d and d′*). These gain- and loss-of-function results together demonstrate that Pk functionally synergizes with Vangl during CE.

We then tested how Pk may affect Dvl function during CE. When co-injected, *mPk2* or *Xpk* can dose-dependently rescue the CE defects induced by *Dvl2* over-expression (*Figure 1e–f′*), suggesting that Pk functionally antagonizes Dvl during CE. Together with our previous finding that Vangl exerts bimodal regulation of Dvl (*Seo et al., 2017*), these results suggest that Pk may synergize with Vangl to suppress Dvl function during CE. Consistent with this idea, we found that the severe CE defects induced by *Pk* and *Vangl2* co-injection could be rescued by overexpressing *Dvl2* (*Figure 1g and g′*). Using activin-induced animal cap elongation as an additional assay for CE, we further confirmed that *Xpk* synergizes with *mVangl2* to induce severe CE defect, which can be rescued by co-overexpression of Dvl2 (*Figure 1h and h*;).

## Vangl interaction with and recruitment of Pk to the plasma membrane is essential for their functional synergy

To understand how Pk synergizes with Vangl to repress Dvl, we first investigated whether they could modulate each other's protein levels. It was reported that Vang could control Pk stability indirectly through ubiquitination in flies (*Cho et al., 2015*; *Strutt et al., 2013*), while in zebrafish, Pk could down-regulate Dsh/Dvl protein level (*Carreira-Barbosa et al., 2003*). In *Xenopus* embryos or explants undergoing CE, however, we found that morpholino knockdown of *Xvangl2* or overexpression of *mVangl2* did not affect the protein level of co-injected mPk2 or XPk (*Figure 2—figure supplement 1a, b, c*). In contrast, co-transfecting XPk with Vangl2 in cultured HEK293T cells did lead to significant reduction of XPk protein level (*Figure 2—figure supplement 1d*, also see Pk2 down-regulation by Vangl2 in 293 cells in *Nagaoka et al., 2019*). Therefore, Vang/Vangl2 modulation of Pk stability seems to be context-dependent and does not account for the observed synergy between Vangl2 and Pk during *Xenopus* CE. Furthermore, overexpression of Pk does not alter the protein level of co-injected Vangl2 or Dvl2 (*Figure 2—figure supplement 1a, b*), indicating that during *Xenopus* CE, Pk does not synergize with Vangl2 or antagonize Dvl2 by altering their protein levels.

To explore other mechanisms that may explain how Pk synergizes with Vangl2 to antagonize Dvl during CE, we examined the effect of Vangl2 on Pk's sub-cellular localization. When EGFP-tagged mPk2 is expressed in either the animal cap or DMZ, it displays diffused cytoplasmic distribution and variable enrichment at the plasma membrane (*Figure 2—figure supplement 2a, c*; *Figure 2—figure supplement 3a*). Co-injection of mVangl2 significantly increased plasma membrane enrichment of mPk2 in both animal cap and DMZ explants (*Figure 2—figure supplement 2b*; *Figure 2—figure supplement 3b, e*). On the other hand, morpholino knockdown of endogenous *Xvangl2* diminished mPk2 plasma membrane enrichment, which could be restored by co-injection of a small amount of mVangl2 (*Figure 2—figure supplement 2d, e*; *Figure 2—figure supplement 3c, d and e*). Together, these data indicate that Vangl2 is both necessary and sufficient to recruit Pk to the plasma membrane.

To test whether plasma membrane recruitment of Pk by Vangl is important for their functional synergy, we took advantage of a Vangl2 R177H variant identified in a patient with diastematomyelia (*Kibar et al., 2011*). This variant changes the highly conserved Arg177 to a histidine in the intracellular loop region between TM2 (transmembrane domain) and TM3 (*Figure 2a*). Importantly, this variant does not perturb Vangl2 plasma membrane trafficking/localization (*Figure 2b and b′*), its protein level (*Figure 2g*, *Figure 2—figure supplement 4e*), or its ability to interact with and recruit Dvl to the plasma membrane (*Figure 2—figure supplement 4a–d*). The variant, however, reduced Vangl2 interaction with Pk and recruitment of Pk to the plasma membrane (*Figure 2c–g*).

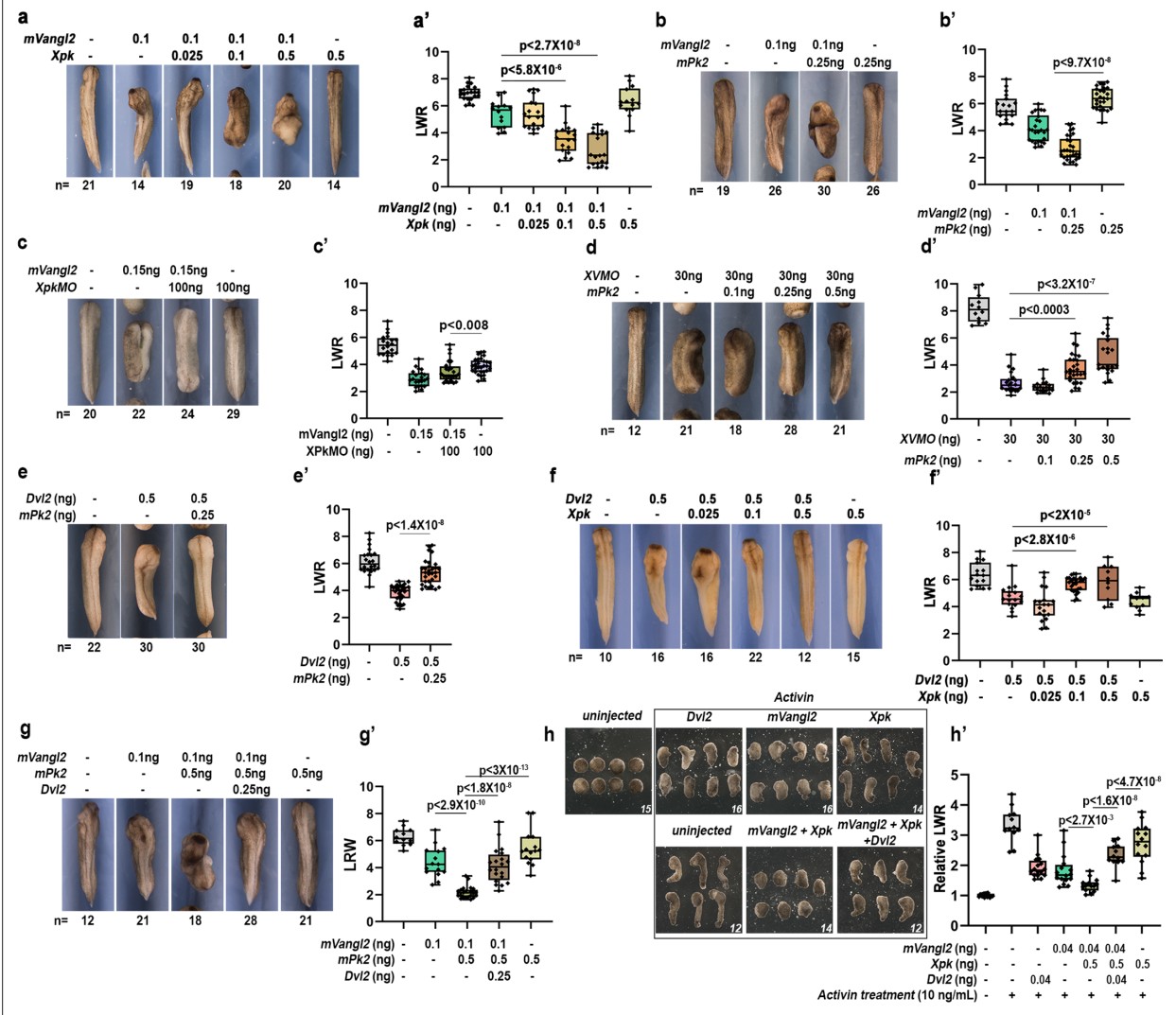

**Figure 1.** Prickle (Pk) synergizes with Vangl to suppress Dishevelled (Dvl) during convergent extension (CE). Injecting 0.1 ng mouse *Vangl2* mRNA (*mVangl2*) into the dorsal marginal zone (DMZ) results in moderate CE defects and reduction of the length-to-width ratio (LWR), and the phenotypes are significantly enhanced by co-injecting a small dose of *Xpk* (**a, a'**) or *mPk2* (**b, b'**) mRNA that causes minimal or no CE defect per se. On the other hand, higher dose *mVangl2* (0.15 ng) induced more severe CE defects that can be rescued by knocking down endogenous Pk using *Xpk*MO (**c, c'**); whereas, knockdown of endogenous *vangl2* (*XVMO*) induced CE defects that can be rescued by moderate mPk2 overexpression (**d, d'**). Conversely, 0.5 ng mouse *Dvl2* injection-induced CE defects can be dose-dependently rescued by *mPk2* (**e, e'**) or *Xpk* (**f, f'**) co-overexpression; and mVangl2-mPk2 co-overexpression-induced severe CE defect can be rescued by co-injecting Dvl2 (**g, g'**). Similarly, in activin-induced animal cap elongation assay, mVangl2 synergizes with Xpk to induce strong CE defect, which can be rescued significantly by co-injection of mouse Dvl2 (**h, h'**). CE phenotype was determined by quantifying the length-to-width ratio (LWR) of the embryos or animal cap explants in each group (**a, b, c, d, e, f, g, h**). Experiments were repeated three times, and the total number of embryos or explants analyzed is indicated below each panel in (**a**)-(**h**). Data are presented as box plots in (**a'**), (**b'**), (**c'**), (**d'**), (**e'**), (**f'**), (**g'**) and (**h'**), with the whiskers indicating the minima and maxima, the center lines representing median, the box upper and lower bounds representing 75th and 25th percentile, respectively. Two-tailed, unpaired T-test was used to compare the LWR of different groups, and the p values are indicated in (**a'**)-(**h'**) between different groups.

The online version of this article includes the following figure supplement(s) for figure 1:

**Figure supplement 1.** Prickle (Pk) knockdown or overexpression induces convergent extension (CE) defects.

Functionally, the R177H variant significantly reduced the synergy between Vangl2 and Pk during CE (***Figure 2h and i***). Moreover, compared to wild-type Vangl2, Vangl2 R177H results in significantly less severe CE defects when over-expressed alone in the DMZ (***Figure 2—figure supplement 5a, a'***), and is less capable of suppressing Dvl-Fz mediated signaling activation during CE when co-expressed (***Figure 2—figure supplement 5b, b'***). We interpret these data to suggest that: (1) direct binding of

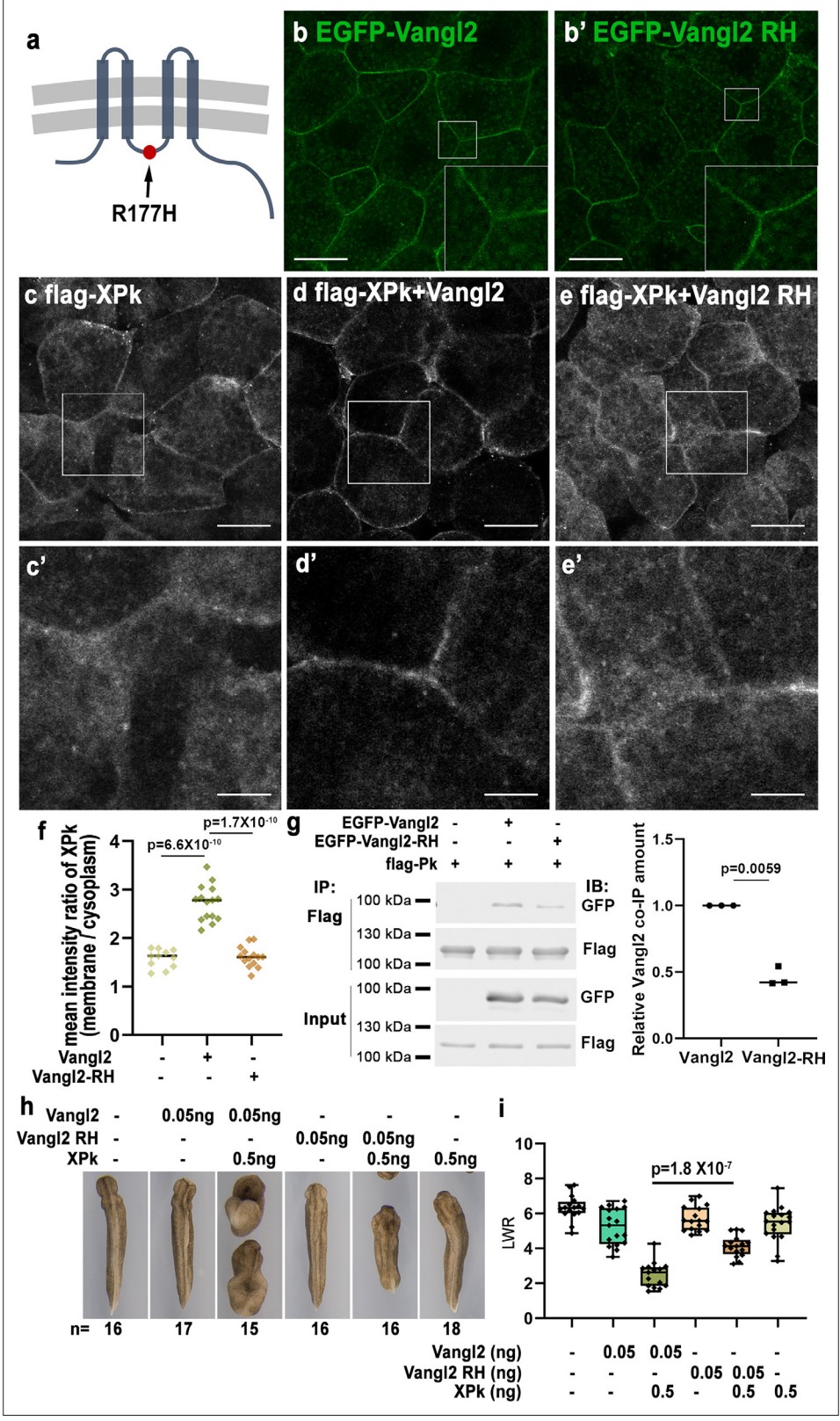

**Figure 2.** Vangl2 RH variant diminishes membrane recruitment of Prickle (Pk) and reduces functional synergy with Pk. Schematic illustration showing the structure of Vangl2 and location of the R177H variant at the intracellular loop region between transmembrane domains 2 and 3 (**a**). When expressed in *Xenopus* animal cap cells, EGFP-Vangl2 RH displays plasma membrane localization indistinguishable from wild-type EGFP-Vangl2 (**b, b'**). Immunostaining

*Figure 2 continued on next page*

*Figure 2 continued*

shows that flag-XPk displays diffuse cytoplasmic localization with some membrane enrichment when expressed alone (**c, c'**). The plasma membrane localization of flag-XPk is enhanced significantly by co-expression of wild-type Vangl2 (**d, d'**), but only modestly by Vangl2 RH variant (**e, e'**). (**f**) Quantification of the ratio of plasma membrane vs. cytoplasmic flag-XPk signal intensity in (**c**), (**d**), and (**e**). Co-IP and western blot show that the R177H mutation does not alter Vangl2 protein level but reduces binding to flag-XPk (**g**, n=3 biological repeats). Functionally, co-injection of 0.05 ng *Vangl2* and 0.5 ng *Xpk* can strongly synergize to disrupt convergent extension (CE), but the synergy is significantly reduced when *Vangl2 RH* variant mRNA is co-injected with *Xpk* (**h**). The CE phenotype was determined by quantifying the length-to-width ratio (LWR) of the embryos in each group in (**h**). Experiments were repeated three times, and the total number of embryos analyzed is indicated below each panel in (**h**). Data are presented as box plots in (**i**), with the whiskers indicating the minima and maxima, the center lines representing median, the box upper and lower bounds representing 75$^{th}$ and 25$^{th}$ percentiles, respectively. Two-tailed, unpaired t-test was used to compare the LWR of different groups, and the p values are indicated between different groups. Scale bars represent 30 μm in b-e; 10 μm in c', d', e'.

The online version of this article includes the following source data and figure supplement(s) for figure 2:

**Source data 1.** PDF file containing original western blots for *Figure 2g*, indicating the relevant bands and treatments.

**Source data 2.** Original files for western blot analysis displayed in *Figure 2g*.

**Figure supplement 1.** Prickle (Pk), Vangl2 and Dvl2 do not cross-regulate each other's protein stability in *Xenopus* embryos during convergent extension (CE).

**Figure supplement 1—source data 1.** PDF file containing original western blots for *Figure 2—figure supplement 1*, indicating the relevant bands and treatments.

**Figure supplement 1—source data 2.** Original files for western blot analysis displayed in *Figure 2—figure supplement 1*.

**Figure supplement 2.** Vangl2 enhances Pk2 plasma membrane localization in animal cap cells.

**Figure supplement 3.** Vangl2 enhances Pk2 plasma membrane localization in dorsal marginal zone (DMZ) cells.

**Figure supplement 4.** Vangl2 R177H variant does not alter Vangl2-Dvl2 interaction.

**Figure supplement 4—source data 1.** PDF file containing original western blots for *Figure 2—figure supplement 4*, indicating the relevant bands.

**Figure supplement 4—source data 2.** Original files for western blot analysis displayed in *Figure 2—figure supplement 4*.

**Figure supplement 5.** R177H mutation diminishes Vangl2 activity in convergent extension (CE).

---

Vangl2 to Pk, through which Pk is recruited to the plasma membrane, is crucial for their functional synergy during CE; and (2) direct binding of Vangl to Dvl alone may not be sufficient to suppress Dvl, and simultaneous interaction with Pk may be required for Vangl to efficiently inhibit Dvl during CE.

## Pk synergizes with Vangl to sequester Dvl from Fz

To investigate how Pk may help Vangl to suppress Dvl during CE, we first tested whether Pk over-expression may disrupt Dvl interaction with Fz. In both animal cap and DMZ cells, Fz7 can recruit mCherry-tagged Dvl2 (Dvl2-mCh) to the plasma membrane (*Figure 3—figure supplement 1b, e*). Co-expression of GFP-mPk2, however, does not perturb Fz7-mediated plasma membrane recruitment of Dvl2 (*Figure 3—figure supplement 1c, f*), suggesting that overexpression of Pk alone cannot effectively disrupt Dvl-Fz interaction in *Xenopus*.

We then investigated the alternative possibility that Pk may help Vangl to sequester Dvl, and thereby preventing Wnt-Fz-Dvl interaction to inhibit non-canonical Wnt signaling activation during CE. Our previous studies provided evidence that Vangl recruits Dvl into an inactive complex at the plasma membrane, and Wnt11 can induce dissociation of Dvl from Vangl (*Angermeier et al., 2025*; *Seo et al., 2017*). We, therefore, first tested whether Pk may reinforce Vangl-Dvl interaction to counter the dissociation effect by Wnt11. When co-injected into the DMZ, mPk2 significantly prevented Wnt11-induced dissociation of Flag-Dvl2 from EGFP-Vangl2, although in the absence of Wnt11 co-injection, it did not substantially increase Dvl2-Vangl2 interaction (*Figure 3—figure supplement 2*). These data suggest that, at least under the condition of our co-IP experiment, Pk may not directly impact the

steady-state binding between Vangl and Dvl, but may strengthen Dvl sequestration by Vangl to inhibit its response to non-canonical Wnt ligand.

To investigate how Pk may help Vangl to regulate Dvl's response to non-canonical Wnt, we performed imaging studies. In *Xenopus* and zebrafish, Wnt11 can induce formation of Fz-Dvl complexes that cluster as patches at the cell-cell contacts (*Angermeier et al., 2025*; *Witzel et al., 2006*; *Yamanaka and Nishida, 2007*). We, therefore, tested how Vangl/Pk may affect formation of Wnt11-induced Fz-Dvl patches. In the DMZ (*Figure 3—figure supplement 3a–c'*) or animal cap (*Figure 3a–c'*) explants, co-injection of *Xenopus Wnt11* can induce Dvl2-EGFP, Dvl2-mScarletI (Dvl2-mSc), or Dvl2-mCh to form distinct patches along cell-cell contacts. By separate injection of *Dvl2-mSc* and *Dvl2-EGFP* into two adjacent blastomeres, we found that the Wnt11-induced Dvl2-mSc and Dvl2-EGFP patches are aligned along the adjacent cell borders (*Figure 3—figure supplement 3a–c'*). This result is consistent with the previous reports (*Witzel et al., 2006*; *Yamanaka and Nishida, 2007*) and indicates that Wnt11 induces symmetric clustering of Dvl across the border of adjacent cells, a scenario that is different from the asymmetric clustering of Fz/Dvl and Vang/Pk complexes between adjacent cells commonly observed during PCP establishment in epithelial tissues.

When EGFP-tagged *Xenopus* Fz7 is co-injected in animal cap or DMZ explants, it exclusively forms membrane patches that completely overlapped with Dvl2 (*Figure 3d–f'*; *Figure 3—figure supplement 3d–f'*), consistent with clustering of Fz with Dvl. Co-injecting a moderate level of EGFP-tagged mouse Vangl2 (0.1 ng) does not perturb Wnt11-induced Dvl2 patch formation. In contrast to Fz7, Vangl2 is distributed broadly on the plasma membrane, but to our surprise, it also displays overlapping enrichment with Dvl2 patches (*Figure 3g–i*; *Figure 3—figure supplement 3g–i*). Close examination revealed that in most cases, Vangl2 is enriched at the edges of Dvl2 patches but diminished at the center (*Figure 3g'-i'*, *Figure 3—figure supplement 3g'-i''*, compare red arrows and arrowheads to green arrows and arrowheads). In 3-D reconstructed confocal images, enriched Vangl2 often forms rings that encircle Dvl2 patches (*Figure 3—figure supplement 4c*, white arrows; and enlarged views in a'-c').

To quantitatively assess their spatial distribution, we measured and plotted the relative intensity of Dvl2 against Fz7, Vangl2, or membrane-GFP control along the length of ten representative patches in animal cap explants (*Figure 3p–r*). Our analyses revealed that membrane-GFP displayed no enrichment along Wnt11-induced Dvl2 patches (*Figure 3p*). Fz7, however, showed enrichment that correlated strongly with Dvl2: their intensities followed a similar pattern of increasing sharply from the edge and peaking coincidentally at the center of the patches (*Figure 3q*). In contrast, Vangl2 enrichment starts to appear slightly outside of Dvl2 patches (*Figure 3g', i' and r*, green arrowheads), peaks at the edges as Dvl2 intensity begins to rise, and dips at the center where Dvl2 and Fz7 intensities reach the maximum (*Figure 3g', i' and r*, green arrows). Analyses of DMZ explants showed the same results (*Figure 3—figure supplement 3q, r*). These imaging analyses are consistent with our co-IP data (*Figure 3—figure supplement 2*) and further suggest the possibility that Dvl may leave Vangl and transition to Fz upon Wnt11 induction.

We then tested whether the addition of Pk can help Vangl2 to counter the effect of Wnt11. Indeed, co-injecting Pk with 0.1 ng Vangl2 effectively reduced Wnt11-induced Dvl2 patch formation, making Dvl2 more evenly distributed along the plasma membrane (*Figure 3m–o*; *Figure 3—figure supplement 3m–o*) and overlap with Vangl2 (*Figure 3m'–o' and t*; *Figure 3—figure supplement 3m'–o, t'*). A similar reduction of Dvl patch formation can also be achieved, albeit less effectively, by over-expression of high-level Vangl2 alone (*Figure 3j–l' and s*; *Figure 3—figure supplement 3j–l, s'*).

To examine how Vangl2 and Pk could affect Fz7 enrichment in the Wnt11-induced Fz/Dvl patches, we co-injected fluorescent protein-tagged Dvl2 and Fz7 in both animal cap and DMZ explants (*Figure 4*; *Figure 4—figure supplement 1*). In both cases, moderate over-expression of Vangl2 or Pk individually does not affect enrichment of Fz7 within Wnt11-induced Dvl2 patches (compare *Figure 4a–c'* with d-i'; *Figure 4—figure supplement 1a–c'* with d-l'), but Vangl2 and Pk together not only perturb Dvl2 patches, but also disperse Fz7 into small puncta (*Figure 4j–l'*; *Figure 4—figure supplement 1j–l'*). Close examination revealed that some of the Fz7 puncta are on the plasma membrane and largely co-localize with Dvl2. The rest of Fz7 puncta, however, are located in the cytoplasm near the plasma membrane and appear to be endocytosed vesicles (arrows in *Figure 4k', l'*, *Figure 4—figure supplement 1k' and l'*). Interestingly, these cytoplasmic puncta contain only Fz7 but not Dvl2 (compare arrows in *Figure 4j' to k'*; and arrows in *Figure 4—figure supplement 1j' to k'*).

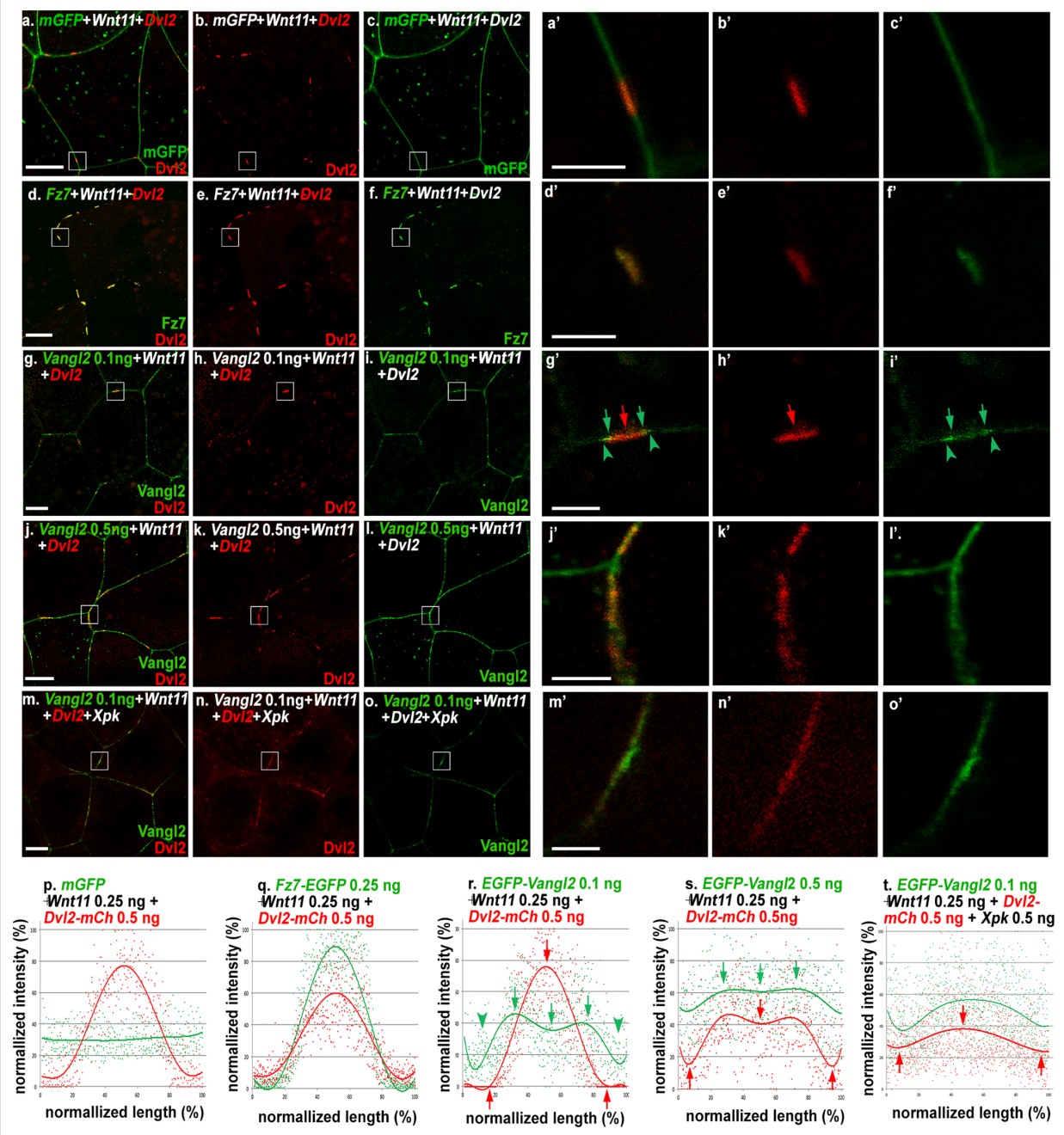

**Figure 3.** Prickle (Pk) synergizes with Vangl2 to inhibit Wnt11-induced formation of Dishevelled (Dvl) patches. In animal cap explants, 0.25 ng *Wnt11* injection with 0.5 ng of mCh-tagged mouse *Dvl2* (*Dvl2-mCh*) and membrane-GFP (*mGFP*) induces formation of distinct Dvl2 patches at the cell-cell contact (**a-c'**). Co-injection indicates that these Dvl2 patches completely overlap with Fz7-EGFP (**d-f'**). In contrast, Vangl2, when expressed at moderate levels (0.1 ng), is distributed more broadly along the plasma membrane (**g–i**), but also displays enrichment immediately outside and at the edge of Dvl2 patches (g'-l', arrowheads and arrows, respectively). A high level of *Vangl2* injection (0.5 ng) inhibits Wnt11-induced Dvl2 patch formation and makes Dvl2 more evenly distributed with Vangl2 (**j-l'**). The same effect can also be achieved by co-expressing Pk with moderate levels of Vangl2 (**m-o'**). (**p–t**) Measurement of the relative intensity of Dvl2 along the patches with either membrane GFP, Fz7, Vangl2 at moderate (0.1 ng) and high (0.5 ng) levels, or Vangl2 (0.1 ng) with *Xpk* (0.5 ng) co-injection. Scale bars represent 15 µm in a-o; 4 µm in a'-o'.

The online version of this article includes the following source data and figure supplement(s) for figure 3:

**Figure supplement 1.** Overexpression of Prickle (Pk) alone cannot effectively disrupt Dvl-Fz interaction in *Xenopus*.

**Figure supplement 2.** Prickle (Pk) inhibits Wnt11-induced dissociation of Dvl2 from Vangl2.

**Figure supplement 2—source data 1.** PDF file containing original western blots for *Figure 3—figure supplement 2*, indicating the relevant bands

*Figure 3 continued*

and treatments.

**Figure supplement 2—source data 2.** Original files for western blot analysis displayed in *Figure 3—figure supplement 2*.

**Figure supplement 3.** Prickle (Pk) synergizes with Vangl2 to inhibit Wnt11-induced formation of Dishevelled (Dvl) patches in dorsal marginal zone (DMZ) explants.

**Figure supplement 4.** Vangl2 forms rings to encircle Wnt11-induced Dvl2 patches on the plasma membrane.

To confirm that the cytoplasmic Fz7 puncta are endocytosed vesicles, we performed FM4-64 dye uptake experiment (*Cho et al., 2015*; *Classen et al., 2005*). FM4-64 is a membrane-impermeable fluorescent dye that can only be internalized through endocytosis. When we incubated the explants with FM4-64, we found that many Fz7 puncta induced by Vangl2/Pk co-injection were also positive for FM4-64 (*Figure 4—figure supplement 2d–f'*), indicating that they are indeed endocytic vesicles.

These data imply that Pk may assist Vangl to sequester Dvl, thereby reducing the accessibility of Dvl to attenuate Fz-Dvl complex formation in response to Wnt11 and resulting in Fz destabilization at the plasma membrane. To test this idea, we reduced Dvl availability at the plasma membrane using another strategy. Over-expression of DshMA, a mitochondrial tethered Dvl, can sequester endogenous Dvl to the mitochondria (and away from the plasma membrane) through DIX-domain mediated oligomerization (*Park et al., 2005*). We found that DshMA injection indeed mimicked the effect of

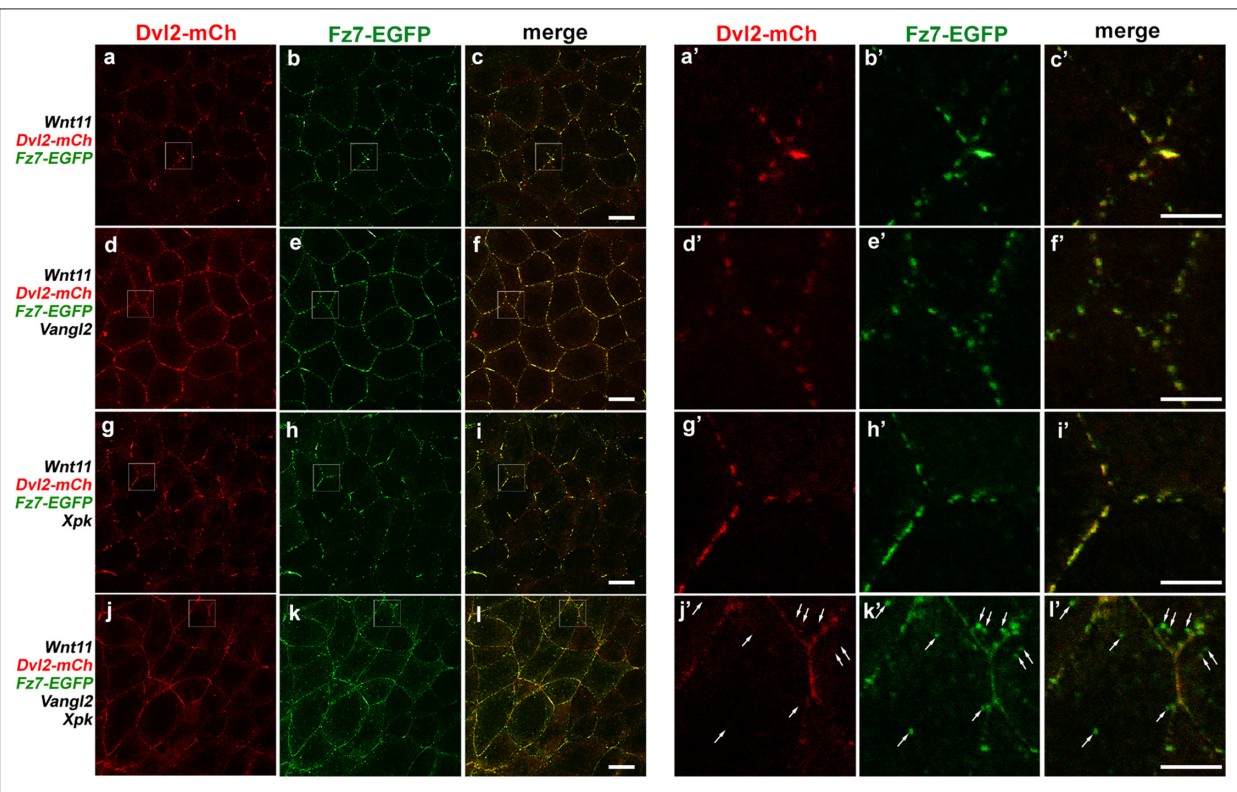

**Figure 4.** Prickle (Pk) helps Vangl2 to inhibit Wnt11-induced clustering of Fz7-Dvl2 complexes. In animal cap explants, Wnt11 induces formation of overlapping Fz7-EGFP and Dvl2-mCh patches at the cell-cell contact (**a-c'**). These patches are not affected by over-expressing moderate levels of Vangl2 (d-f', 0.1 ng mRNA) or XPk (g-i', 0.5 ng mRNA) individually. Vangl2 and XPk co-expression, however, not only disrupts Dvl2-mCh patches but also disperses Fz7-EGFP patches into small puncta (**j–l**). Enlarged views revealed that some of the Fz7-EGFP puncta are on the plasma membrane and remain co-localized with Dvl2-mCh, while the others are located in the cytoplasm near the plasma membrane (k″, l', arrows) and contain only Fz7 but not Dvl2 (compare arrows in j' to k'). Scale bars represent 30 μm in a-l; 10 μm in a'-l'.

The online version of this article includes the following figure supplement(s) for figure 4:

**Figure supplement 1.** Prickle (Pk) helps Vangl2 to inhibit Wnt11-induced clustering of Fz7-Dvl2 complexes in dorsal marginal zone (DMZ) explants.

**Figure supplement 2.** Prickle (Pk)/Vangl2 co-expression induces Fz7 endocytosis.

**Figure supplement 3.** Mitochondria sequestration of endogenous Dishevelled (Dvl) induces Fz7 endocytosis.

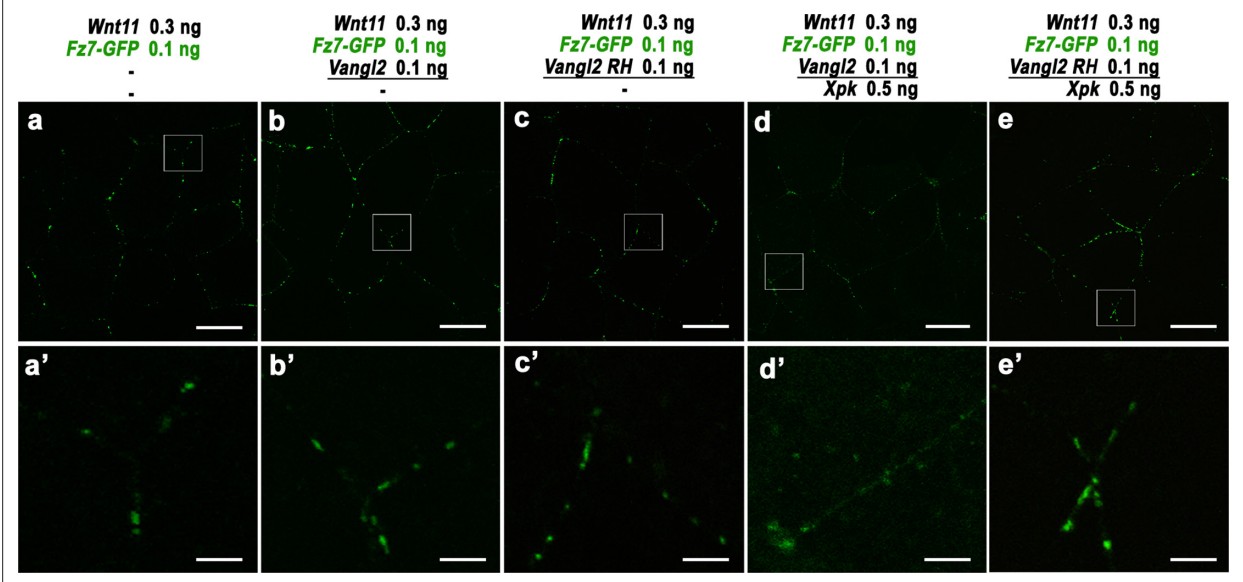

**Figure 5.** Vangl2 R177H variant fails to synergize with Prickle (Pk) to inhibit Fz patch formation and down-regulate Fz stability at the plasma membrane. Wnt11-induced formation of Fz7-GFP patches on the plasma membrane (**a, a'**) was not affected by moderate expression either of wild-type Vangl2 (**b, b'**) or Vangl2 R177H variant (**c, c'**) alone. *Xpk* co-injection synergized with wild-type Vangl2 to diminish Fz7 patch formation and induce cytoplasmic Fz7 puncta (**d, d'**), but the synergy was not observed with Vangl2 R177H (**e, e'**). Scale bars represent 30 μm in a-e; 5 μm in a'-e'.

Vangl2/Pk co-injection on Fz, resulting in reduced Fz7 clustering upon Wnt11 induction, formation of cytoplasmic puncta near the plasma membrane, and diminished plasma membrane localization (*Figure 4—figure supplement 3*).

To further test whether Vangl2 needs direct interaction with Pk in order to down-regulate Fz7 stability at the plasma membrane and Fz7 patch formation in response to Wnt11, we analyzed the Vangl2 R177H variant that specifically reduces Vangl2-Pk interaction (*Figure 2*). We found that, unlike wild-type Vangl2, co-injecting Vangl2 R177H with Pk failed to significantly diminish Wnt11-induced Fz7 patch formation or cause cytoplasmic Fz7 puncta (*Figure 5*). These data support the notion that direct Pk-Vangl2 interaction is required for efficient sequestration of Dvl from Fz, thereby reducing Fz stability on the plasma membrane.

As a final test for this idea, we examined Dvl phosphorylation known to be inducible by Fz and non-canonical Wnt signaling activation (*Axelrod, 2001*; *Klein et al., 2006*; *Rothbächer et al., 2000*; *Shimada et al., 2001*; *Strutt et al., 2019*; *Strutt et al., 2006*). *Xenopus* extract from embryos injected with *flag-Dvl2* in the DMZ often shows Dvl2 migrating as two bands. The upper, slower-migrating band increases in intensity from stage 10–12, correlating with the onset and progression of CE (*Figure 6a*). The slower migrating form of Dvl2 is increased by Fz7 co-injection but eliminated by phosphatase treatment (*Figure 6b*). Conversely, high-level Vangl2 overexpression reduces the phosphorylated form of Dvl2, while Fz7 can counter Vangl2's effect to increase Dvl2 phosphorylation when co-injected (*Figure 6c*). Furthermore, our co-IP experiment demonstrated that only the faster migrating, presumably unphosphorylated form of Dvl2 could be pulled down by Vangl2 in *Xenopus* (*Figure 6d*), suggesting that Vangl2-bound Dvl2 may be shielded from Fz-induced phosphorylation. High-level Pk (1 ng *Xpk*) is not sufficient to significantly reduce phosphorylated form of Dvl2 when injected alone, but can synergize with moderate level of co-injected Vangl2 to reduce Dvl2 phosphorylation (*Figure 6e and f*). Together with our previous findings, these data suggest that Pk facilitates Vangl2 to sequester Dvl2 from Fz and, in turn, Fz-induced phosphorylation.

## Ror2 facilitates the transition of Dvl2 from Vangl2 to Fz complexes in response to non-canonical Wnt

The above results prompted us to ask that if Dvl is sequestered at the plasma membrane by Vangl2/Pk, how it may transition to form a complex with Fz in response to non-canonical Wnt? As Ror2 has been shown to act as a non-canonical Wnt co-receptor capable of interacting with both Fz and Vangl2

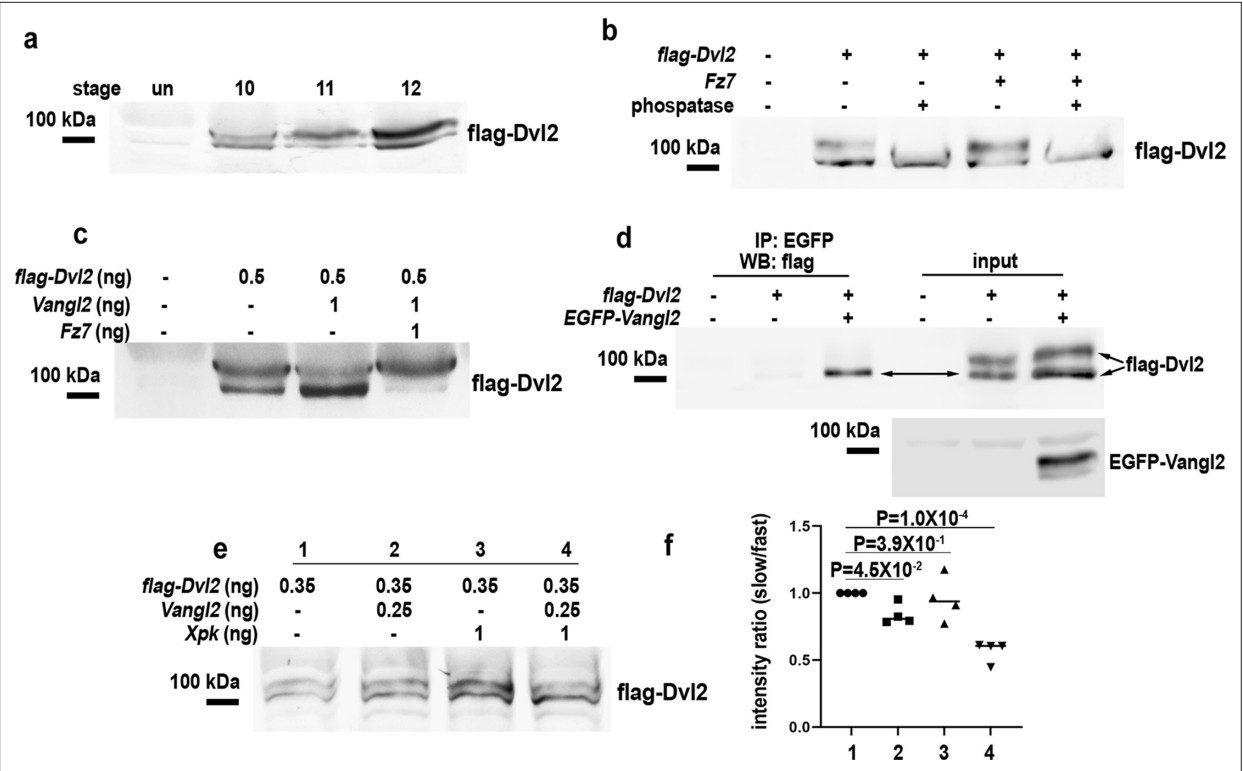

**Figure 6.** Prickle (Pk) synergizes with Vangl2 to prevent Frizzled (Fz)-induced phosphorylation of Dishevelled (Dvl). In *Xenopus* extract, dorsal marginal zone (DMZ)-injected Dvl2 migrates as two bands, with the slower migrating band increasing in intensity from stage 10–12 as convergent extension (CE) starts and progresses during gastrulation (**a**). The slower migrating form of Dvl2 is increased by Fz7 co-injection but eliminated by phosphatase treatment (**b**). High-level Vangl2 injection can reduce the phosphorylated form of Dvl2, while Fz7 can counter Vangl2's effect to increase Dvl2 phosphorylation (**c**). Co-IP experiment indicates that only the faster migrating, presumably unphosphorylated form of Dvl2 can be co-immunoprecipitated by Vangl2 (**d**). High-level *Xpk* (1 ng) or moderate *Vangl2* (0.25 ng) cannot significantly reduce phosphorylated form of Dvl2 when injected individually, but their co-injection can suppress Dvl2 phosphorylation (**e**). (**f**) Quantification of the ratio between the slow-migrating/ phosphorylated and fast-migrating/unphosphorylated forms of Dvl2 in (**e**), n=3 biological repeats.

The online version of this article includes the following source data for figure 6:

**Source data 1.** PDF file containing original western blots for **Figure 6**, indicating the relevant bands and treatments.

**Source data 2.** Original files for western blot analysis displayed in **Figure 6**.

during CE (*Gao et al., 2011*; *Grumolato et al., 2010*; *Hikasa et al., 2002*; *Ho et al., 2012*; *Wallkamm et al., 2014*), we hypothesized that Ror2 may be a key component to shuttle Dvl between Vangl2 and Fz.

To test this idea, we first examined the functional relationship between Ror2 and Vangl2. Injecting a moderate amount of *Xenopus ror2* mRNA (*Xror2*; 0.05–0.1 ng) can efficiently rescue the severe CE defects induced by 0.2 ng of *Vangl2* mRNA (**Figure 7—figure supplement 1a, b**), supporting the idea that, like Dvl2, Ror2 antagonizes Vangl2 to activate non-canonical Wnt signaling during CE.

Second, we tested at the cellular level how Wnt11 may induce Ror2 to cluster into patches and how Ror2 patches may correlate with Dvl2 and/or Vangl2 patches. Similar to Fz and Dvl, Ror2-EGFP can be induced to form patches on the plasma membrane by Wnt11 (**Figure 7a–f**). Upon co-injection with Dvl2-mCh, the Ror2 patches are overlapped with Dvl2 patches (**Figure 7g–i**). Close examination of these patches revealed that Ror2, like Fz, accumulates with Dvl2 to high levels in the center of the patches (red arrowhead in **Figure 7g' and j**). But unlike Fz, Ror2 patches are slightly longer and extend outside of Dvl2 patches (green arrowheads in **Figure 7g' and j**). Quantification indicated that the signal intensity ratio between Ror2 and Dvl2 is increased over twofolds at the border of the patches (**Figure 7j**, bottom panel). This is reminiscent of Vangl2 enrichment at this region (**Figure 3g'–i'; r**), and suggests that Ror2 and Vangl2 may accumulate together at the border of Dvl2 patches to form a complex with reduced amount of Dvl2. Also similar to Vangl2, Ror2 continues to display broad

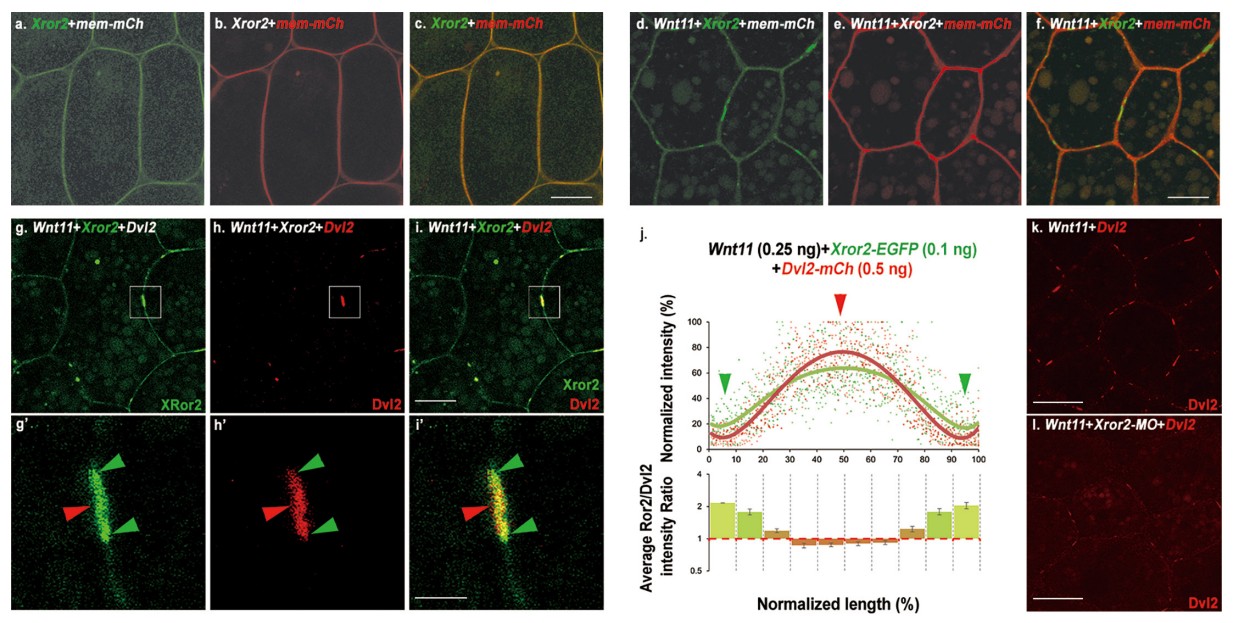

**Figure 7.** Ror2 is an obligatory component of the Frizzled (Fz)/Dishevelled (Dvl) cluster complex induced by Wnt11. EGFP-tagged *Xenopus* Ror2 (0.1 ng mRNA injection) is distributed homogeneously on the plasma membrane when injected with *membrane-mCherry mem-mCh* (**a–c**), but can be induced to form distinct patches upon *Wnt11* co-injection (**d-f**, 0.25 ng). With Dvl2-mCh co-injection (0.5 ng), the Ror2 patches show overlap with Dvl2 patches (**g–i**). Unlike Dvl2, however, Ror2 additionally displays broad distribution along the entire cell cortex (compare g to h, and also see **d**). The enlarged views show that both Ror2 and Dvl2 accumulate at the center of the patches (red arrowhead), but Ror2 patches are slightly longer and extend beyond the border of Dvl2 patches (green arrowheads in g'-i'). (**j**) Measurement of the relative intensity of Dvl2-mCh along the patches with Ror2-EGFP (upper panel), and quantification of the ratio between Ror2 and Dvl2 intensity along the patches (bottom panel). Wnt11 induces Dvl2-mCh patch formation on the cell cortex (**k**) and is blocked by co-injecting 25 ng *Xror2* morpholino (*Xror2*-MO) (**l**). Scale bars represent 15 μm in a-i; 4 μm in g'-i'; 30 μm in k-l.

The online version of this article includes the following source data and figure supplement(s) for figure 7:

**Figure supplement 1.** Ror2 co-overexpression rescues Vangl2 overexpression-induced convergent extension (CE) defects.

**Figure supplement 2.** In *Xenopus* embryo extract, the co-IP experiment shows that binding between Dvl2-EGFP and Myc-Vangl2 is reduced by Wnt11 (compare lanes 2 and 3).

**Figure supplement 2—source data 1.** PDF file containing original western blots for *Figure 7—figure supplement 2*, indicating the relevant bands and treatments.

**Figure supplement 2—source data 2.** Original files for western blot analysis displayed in *Figure 7—figure supplement 2*.

**Figure supplement 3.** XRor2 knockdown does not alter plasma membrane recruitment of Dvl2 by Vangl2.

**Figure supplement 4.** Ror2 overexpression alone cannot recruit Dvl2 to the plasma membrane.

membrane distribution outside of Dvl2 patches (*Figure 7d and g*) in the presence of Wnt11, differing from Dvl2 and Fz that are localized exclusively within patches (*Figure 7h and i*; *Figure 3d–f*). These results suggest that at least under the moderate over-expression condition for our imaging experiments, a portion of Ror2 may remain tethered to Vangl2 on the plasma membrane, whereas most Dvl2 dissociates from Vangl2 to cluster with Fz in response to Wnt11.

To test whether Ror2 is required for Dvl2 patch formation in response to Wnt11, we used a verified morpholino to knock down endogenous XRor2 (*Schambony and Wedlich, 2007*) and found that Dvl2 patch formation is significantly diminished in Xror2 morphants (*Figure 7k and l*). Collectively, these data indicate that Ror2, a molecular partner of Vangl2, is an obligatory component of the Fz/Dvl cluster induced by Wnt11.

To further scrutinize the molecular mechanism, we tested biochemically whether Ror2 is required for Dvl2 to dissociate from Vangl2 in response to Wnt11. Unexpectedly, the co-IP data showed that the steady-state binding between Dvl2 and Vangl2 appeared to be reduced by XRor2 knockdown (*Figure 7—figure supplement 2a, a'*). The reason for this is unclear, since our imaging showed normal Dvl2 plasma membrane recruitment by Vangl2 with Ror2 knockdown (*Figure 7—figure supplement 3*). Irrespective of the reason for the decreased Dvl2-Vangl2 co-IP with XRor2 knockdown, co-injection

of Wnt11 cannot further reduce Dvl2-Vangl2 binding in the absence of Ror2 (*Figure 7—figure supplement 2a, a'*).

Together, these data support the notion that Ror2 is required for Dvl2 to transition from Vangl2 to Fz in response to Wnt11. To test how Ror2 may facilitate this transition, we performed co-IP and imaging experiments. Our imaging data showed that, unlike Vangl2 and Fz7, Ror2 cannot recruit co-injected Dvl2 to the plasma membrane (*Figure 7—figure supplement 4*). Also similar to a previous report (*Gao et al., 2011*), our co-IP experiment detected Ror2 interaction with Vangl2 but not Dvl2 (*Figure 7—figure supplement 2b*), suggesting that Ror2 may bind directly to Vangl2 but not Dvl2. As Ror2 and Dvl2 can both bind to Vangl2, we reasoned that they could interact indirectly through their mutual binding with Vangl2. We thus performed fluorescence-detection size exclusion chromatography (FSEC) with protein extract from *Xenopus* embryos injected with *Xror2-EGFP*, *HA-Vangl2*, and *Flag-Dvl2*. The elution of Ror2-EGFP was monitored by a fluorescence detector following size exclusion chromatography. The fractions of different molecular sizes were collected and analyzed by Western blot. We found co-fractionation of Ror2, Vangl2 and Dvl2 in fractions 14, 15, and 16 (with the approximate molecular weight of 773–1717, 348–773, and 166–348 kD, respectively; *Figure 7—figure supplement 2c*). This result supports our hypothesis that Ror2, Vangl2, and Dvl2 form complexes in

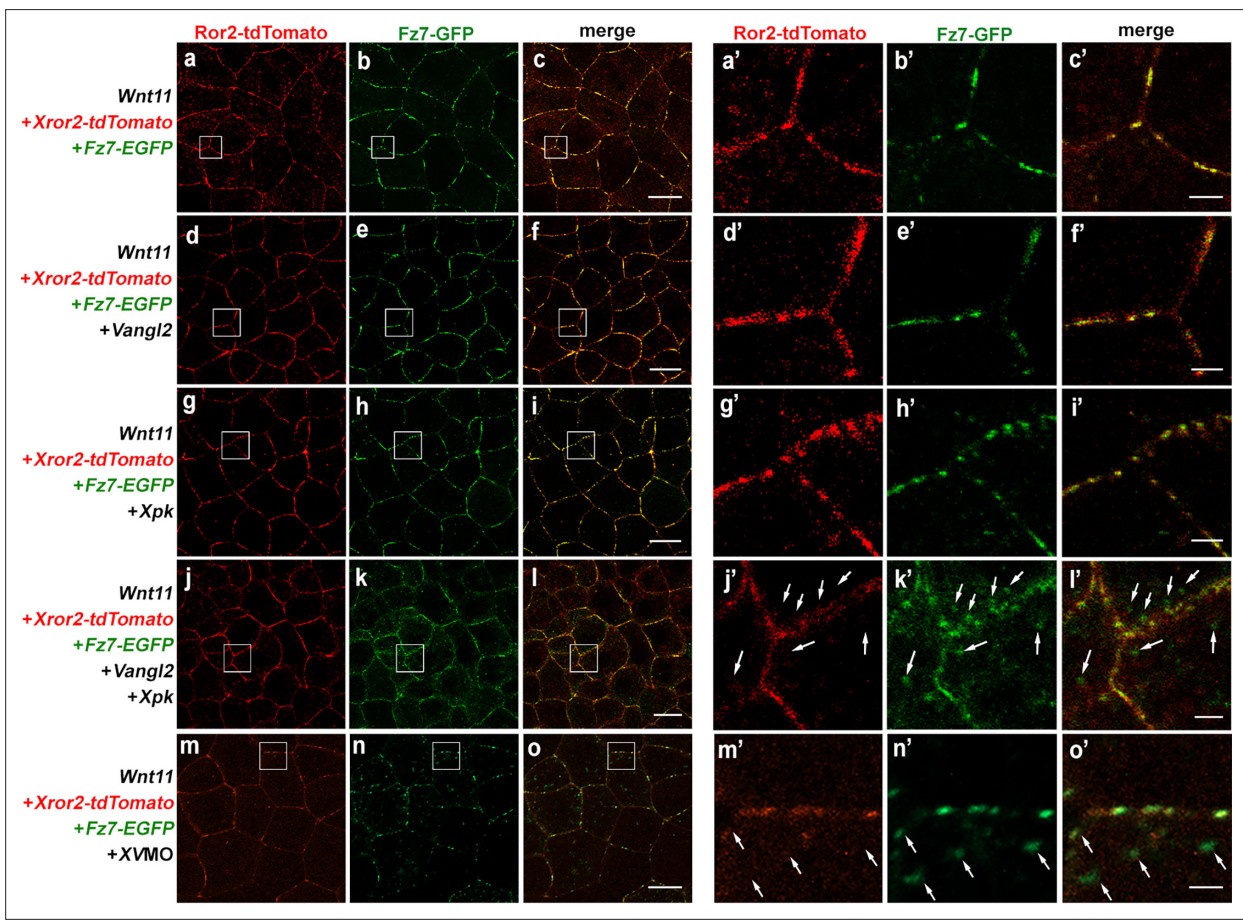

**Figure 8.** Vangl2/Prickle (Pk) exerts bimodal regulation of Ror2 in non-canonical Wnt signaling. (**a–c**) In animal cap explants, Wnt11 induces co-injected XRor2-tdTomato and Fz7-EGFP to co-cluster into patches on the cell cortex. Overexpression of a moderate level of Vangl2 (**d-f**, 0.1 ng) or Pk (**g-l**, 0.5 ng) individually does not perturb co-clustering of Ror2 with Fz7 into patches in response to Wnt11, but their co-overexpression (**j–l**) diminishes Ror2-Fz7 patches into small puncta and causes Fz7 to form cytoplasmic puncta, whereas Ror2 remains on the plasma membrane (compare arrows in **j'-l'**). **a'-l'** are enlarged views of **a-l**, respectively. Conversely, partial knockdown of endogenous *XVangl2* with a moderate level of *XVMO* (14 ng) also diminished Ror2/Fz7 patch formation in response to Wnt11 (**m–o'**) with simultaneous formation of intracellular puncta around the plasma membrane which contain Fz7 but not Ror2 (**m'–o'**), arrows. Scale bars represent 30 µm in a-o; 4 µm in **a'-o'**.

The online version of this article includes the following figure supplement(s) for figure 8:

**Figure supplement 1.** Prickle (Pk) helps Vangl2 to inhibit Wnt11-induced clustering of Fz7 and Ror2.

vivo. We envision a model in which an Ror2/Vangl2/Dvl2 complex serves two purposes during CE: it allows Vangl2 to simultaneously sequester both Ror2 and Dvl2 and keeps them inactive, while in response to non-canonical Wnt, it enables Ror2 to shuttle Dvl to Fz (See Figure 9 and Discussion below).

## Bimodal regulation of Ror2 by Vangl2/Pk during non-canonical Wnt signaling

We used Wnt11-induced patch formation as a readout to test the above model. First, we tested whether Vangl2 may synergize with Pk to sequester Ror2 from Fz7 as it does to Dvl2 (*Figures 3 and 4*). When td-Tomato tagged Ror2 and EGFP tagged Fz7 are co-expressed with Wnt11 in either animal cap (*Figure 8*) or DMZ (*Figure 8—figure supplement 1*) explants, they form clusters that overlap. Moderate overexpression of Vangl2 or XPk individually does not affect Ror2/Fz7 co-clustering within Wnt11-induced patches (compare *Figure 8a–c′ to d-i′*, *Figure 8—figure supplement 1a–c′ to d-i′*). Vangl2 and Pk co-injection, however, significantly reduced Ror2/Fz7 patches induced by Wnt11 into small puncta (*Figure 8j–l*; *Figure 8—figure supplement 1j–l*), and caused Fz7 to form intracellular puncta near the plasma membrane (*Figure 8k′ and l′*, arrows; *Figure 8—figure supplement 1k′, l′*). Interestingly, like Dvl2 (*Figure 4j′ and k′*), Ror2 is not present in these Fz7 puncta (compare arrows in *Figure 8j′ to k′* and in *Figure 8—figure supplement 1j′–k′*) but remained on the plasma membrane, presumably with Dvl2 and Vangl.

We then tested whether Vangl2 may also be required for Ror2 to form Wnt11-induced patches with Fz7 since our model predicts that, by bridging Ror2 and Dvl into a complex, Vangl2 helps Ror2 to shuttle Dvl to Fz in response to non-canonical Wnt (*Figure 9*). We found that partial knockdown of endogenous *XVangl2* with *XV*MO indeed reduced Ror2/Fz7 patches formed in response to Wnt11 (*Figure 8m–o′*) with simultaneous formation of Fz7 intracellular puncta around the plasma membrane (*Figure 8m′ and n′*, arrows), similar to co-overexpression of Vangl2 and XPk (*Figure 8k′ and l′*). Collectively, these data support our model and suggest that with Pk, Vangl2 exerts bimodal regulation of Ror2 in non-canonical Wnt signaling.

## Discussion

Early fly studies identified six proteins that act as core members to coordinate cellular polarity across the plane of the epithelium. In-depth genetic, biochemical, and imaging studies have subsequently elucidated how the six core PCP proteins interact within and between cells to establish feedback loops that partition Vang/Pk and Fz/Dsh/Dgo clusters on opposing cell cortexes to coordinate polarity (*Amonlirdviman et al., 2005*; *Goodrich and Strutt, 2011*). These studies establish a foundation to understand the action of PCP proteins in static epithelial cells. They do not, however, seem to provide direct explanation for how PCP proteins regulate polarized and dynamic cell behavior during CE, where asymmetric partitioning of core PCP proteins has not been consistently observed and non-core PCP proteins, including non-canonical Wnt ligands, co-receptors Ror1/2, and a cytoplasmic protein Dact1, are also critically involved. Furthermore, adopting core PCP proteins to regulate CE is likely a vertebrate-specific adaptation during evolution, since fly germband extension, a CE-like morphogenetic event, does not involve core PCP proteins (*Zallen and Wieschaus, 2004*). We previously provided evidence for a model that during CE, Vangl exerts bimodal regulation of Dvl by cell-autonomously recruiting Dvl to the plasma membrane in an inactive state, and simultaneously poising Dvl for activation upon binding of Fz to non-canonical Wnt ligands (*Seo et al., 2017*). Our recent work tested this model by studying how Dact1, a vertebrate-specific protein, modulates Dvl-Vangl interaction during non-canonical Wnt signaling and CE (*Angermeier et al., 2025*). In the current study, we further tested this model and demonstrated that Pk functionally synergizes with Vangl2 to inhibit Dvl2 during CE in *Xenopus*. Mechanistically, Pk binding to Vangl2 helps Vangl2 to sequester Dvl and constrain its transition to Fz. Moreover, Pk seems to play a similar role in assisting Vangl2 to sequester Ror2, whereas Ror2 is required for Dvl2 to transition from Vangl to Fz in response to non-canonical Wnt. We propose an updated model for the bimodal regulation in which Vangl2/Pk bring both Dvl2 and Ror2 into an inactive complex that prevents ectopic non-canonical Wnt signaling. On the other hand, this complex can also be coupled with Fz upon non-canonical Wnt-induced binding between Ror2 and Fz, delivering Dvl to Fz to initiate non-canonical Wnt signaling

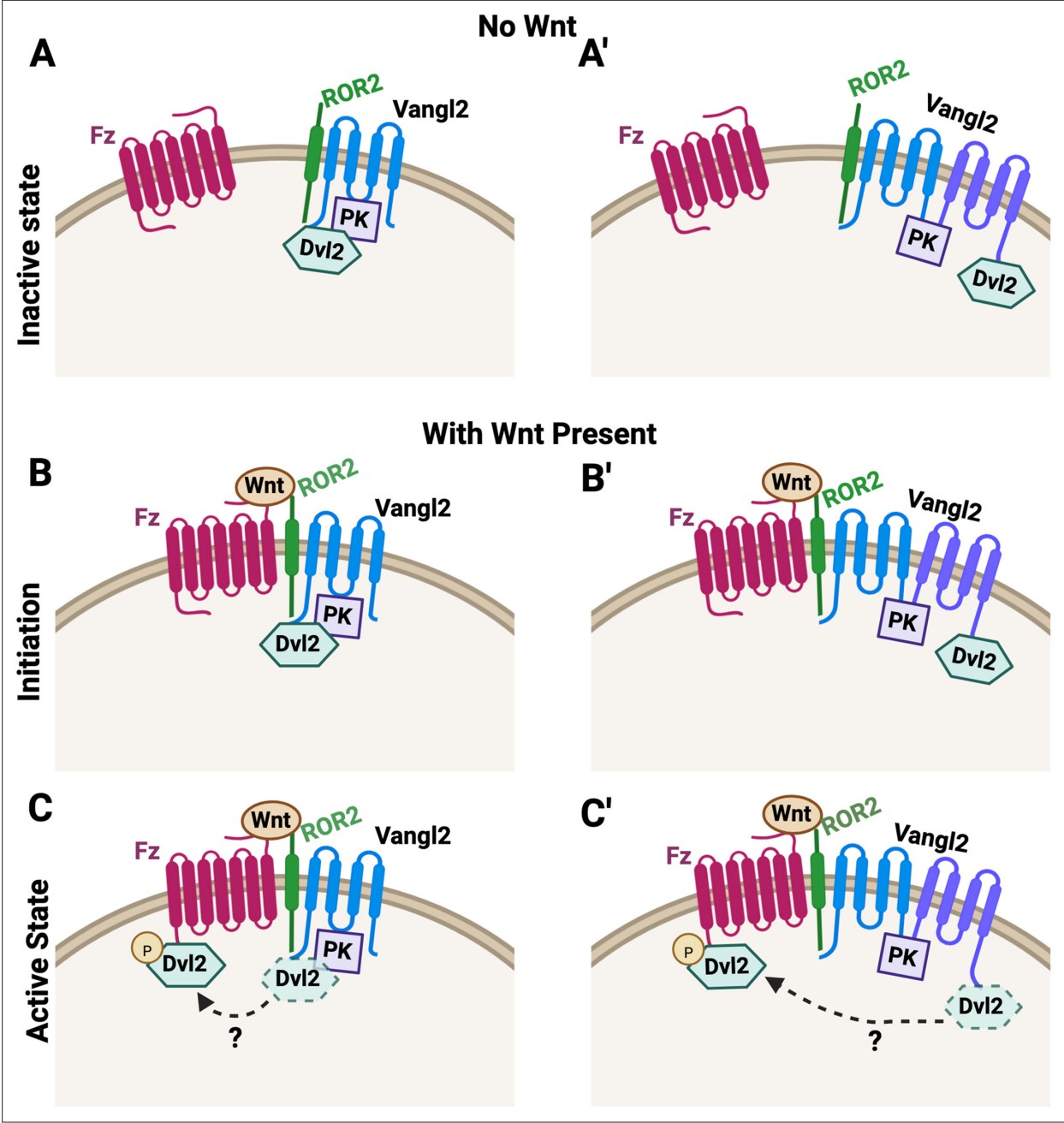

**Figure 9.** An integrated model for non-canonical Wnt signaling regulation during convergent extension (CE). (**a, a'**) In the absence of non-canonical Wnt, Pk helps Van Gogh-like (Vangl) to act as an adaptor that brings together Dishevelled (Dvl) and Ror, through either simultaneous binding of both Dvl and Ror to a single Vangl (**a**) or oligomerization of Vangl proteins bound separately to Dvl and Ror (**a'**), and keeps both Dvl and Ror inactive to prevent ectopic non-canonical Wnt signaling. (**b, b'**) Non-canonical Wnt initiates signaling by triggering Frizzled (Fz)-Ror heterodimerization, and in turn, the complexes consisting of Ror/Vangl/Dvl are brought close to Fz to deliver Dvl for activation of downstream targets. (**c, c'**) Non-canonical Wnt also induces other events, such as Dvl phosphorylation to facilitate Dvl dissociation from Vangl and transition to Fz in a spatially and temporally controlled manner.

(*Figure 9*). Therefore, Vangl-mediated plasma membrane recruitment of Dvl and pre-assembly of Dvl/ Vangl/ Ror complex can also accelerate non-canonical Wnt signaling activation by facilitating Dvl presentation to ligand-bound Fz. Our model provides a new framework to decipher how core PCP proteins are integrated with non-core proteins to tightly control the threshold and dynamics of non-canonical Wnt/ PCP signaling during CE.

## Regulation of Vangl-Dvl interaction by Pk to suppress non-canonical Wnt signaling

Our previous work proposed that Vangl-Dvl interaction provides a key switch to the central logic of non-canonical Wnt signaling by enriching Dvl around the plasma membrane for effective access to Fz, while at the same time keeping Dvl inactive to prevent ectopic signaling (*Seo et al., 2017*). In the current study, we found that the Vangl2 R177H variant (*Kibar et al., 2011*), which can traffic properly and bind to and recruit Dvl2 to the plasma membrane like wild-type Vangl2 (*Figure 2*, *Figure 2—figure supplement 4*), is less capable of inhibiting CE and rescuing Fz/Dvl over-expression induced CE defect (*Figure 2—figure supplement 5*). The results suggest that binding to Dvl per se is not sufficient for Vangl2 to suppress Dvl during CE. Interestingly, Vangl2 R177H displays significantly reduced binding and functional synergy with Pk. We, therefore, reason that interaction with Pk is necessary for Vangl2 to efficiently sequester Dvl from Fz or downstream targets like Daam1. Our biochemical and imaging experiments provide supporting evidence to this idea (*Figures 3–5*; *Figure 3—figure supplements 2 and 3*; *Figure 4—figure supplement 1*; *Figure 8—figure supplement 1*).

There are at least three possibilities accounting for how Pk can assist Vangl2 to sequester Dvl. First, given that Dvl, Pk, and Vang/Vangl can mutually interact with each other (*Bastock et al., 2003*; *Humphries et al., 2023*; *Jenny et al., 2003*; *Takeuchi et al., 2003*; *Tree et al., 2002*), Pk may stabilize a ternary Dvl/Pk/Vangl complex by simultaneously interacting with both Vangl and Dvl (*Figure 9a*). We, however, do not favor this possibility because (1) the binding between Pk and Dvl was reported to be quite weak (*Bastock et al., 2003*); (2) our unpublished data show that ΔPL, a Pk mutant lacking the PET/LIM domains necessary for Dvl binding (*Takeuchi et al., 2003*), can largely mimic wild-type Pk function. Furthermore, a recent study in flies showed that the phosphorylation status of a conserved tyrosine in the cytoplasmic tail of Vang provides opposite binding preference for Pk and Dsh, suggesting that simultaneous binding of both Pk and Dsh to the C-terminus of Vang may not be possible (*Humphries et al., 2023*).

Second, Pk may regulate biochemical modification on Vangl2 to strengthen Vangl2-Dvl interaction. The best-known modification on Vang/Vangl is phosphorylation at several N-terminal serine/threonine residues in response to Wnt/Fz (*Gao et al., 2011*; *Kelly et al., 2016*; *Yang et al., 2017*). Whereas one study in flies reported that Pk can prevent Vang phosphorylation at these residues to decrease Vang turnover (*Strutt et al., 2019*), another study in fly S2 cells found that Pk overexpression does not alter Vang N-terminal phosphorylation (*Kelly et al., 2016*). In our current studies in *Xenopus,* we have not been able to detect Vangl2 phosphorylation consistently, but this possibility remains an interesting idea and should be tested in the future using the reported phosphomutant and phosphomimetic Vangl2 (*Yang et al., 2017*). Finally, phosphorylation of a conserved C-terminal tyrosine residue of Vang was recently reported to decrease its binding with Dsh (*Humphries et al., 2023*) and could, therefore, provide a mechanism to control Vangl-Dvl interaction. But the regulatory mechanism of this tyrosine phosphorylation is not clear and does not seem to depend on Fz. Further studies will be needed to elucidate the role of this tyrosine phosphorylation in vertebrate CE.

Thirdly, Pk binding may induce allosteric change or clustering of Vangl to increase the overall avidity for Dvl binding. Vang/Vangl was proposed to dimerize, and possibly oligomerize into larger clusters through their C-terminal tail and/or transmembrane domains (*Belotti et al., 2012*; *Jenny et al., 2003*), and two recent cryo-EM studies revealed that Vangl1/2 can oligomerize into trimers (*Song et al., 2025*; *Zhang et al., 2025*). Interestingly, quantitative imaging studies in flies have revealed that in stable PCP clusters, the ratio between Vang and Pk is 6:1 (*Strutt et al., 2016*). In light of our data suggesting that the intracellular loop between TM2 and 3 in Vangl2 may impact Pk binding in addition to the canonical Pk binding domain at the C-terminal tail (*Figure 2*; *Bastock et al., 2003*; *Humphries et al., 2023*; *Jenny et al., 2003*), it is tempting to speculate that Pk may nucleate or stabilize Vangl oligomer formation through multimeric interactions with different domains on multiple Vangl proteins. Such oligomeric Vangl cluster may form a 'cage' to more effectively sequester Dvl due to increased local concentration and/or higher binding affinity resulting from conformational change upon oligomerization or Pk binding.

Our above model seemingly contradicts the fly studies showing that Vang/Pk clusters are partitioned to the opposite cell cortexes from Fz/Dsh clusters and are clearly devoid of Dsh. These segregated clusters, however, seem to form progressively from initial symmetrically distributed PCP proteins along cell-cell junctions at early stages where Vang does co-mingle with Dsh (*Bastock et al.,*

*2003*), and a new study further implicated the functional importance of Vang-Dsh binding in fly PCP establishment (*Humphries et al., 2023*). Persistent contact and stable junctions between neighboring cells may facilitate feedback interaction to partition Vang/Pk from Dsh/Fz (*Stahley et al., 2021*). In dynamically moving cells during CE, intercellular feedback interactions are likely limited and transient, therefore, posing challenges for stable segregation of distinct PCP clusters. Conversely, non-canonical Wnt ligands play a key role during vertebrate CE but not in fly PCP establishment. These differences may lead to some changes in the molecular actions of core PCP proteins (see below).

## An Ror-dependent relay mechanism to deliver Dvl for non-canonical Wnt signaling

The premise of our model is that during CE, Vangl acts via a relay mechanism to first bring Dvl to the plasma membrane, and then releases Dvl to Fz. Pk may tighten up this relay mechanism, via regulating Vangl-Dvl interaction, to increase the efficiency of Dvl plasma membrane recruitment and the threshold at which Dvl can be released to Fz. While the detailed mechanisms for how Dvl can be released from Vangl and transitioned to Fz are yet to be elucidated in further details in the future, our studies identified several factors that contribute to the transition: non-canonical Wnt, the co-receptor Ror2, and Dvl phosphorylation.

Our co-IP and imaging studies showed that Wnt11 can trigger dissociation of Dvl2 from Vangl2 (*Figure 3g–i'*; *Figure 3—figure supplements 2 and 3g–i'*; *Angermeier et al., 2025*; *Seo et al., 2017*) and formation of Fz7-Dvl2 clusters at cell-cell contact (*Figure 3d-f'*, *Figure 3—figure supplement 3d-f'*), indicating that non-canonical Wnt can act extracellularly to trigger the transition of Dvl from Vangl to Fz. It is possible that Wnt binding to Fz can directly induce events in favor of Fz-Dvl association. We, however, also consider the alternate possibility that Wnt binding to the co-receptors Ror1/2 brings Dvl to Fz.

Like Fz, Ror1/2 also harbor the extracellular cysteine-rich domains known to interact with Wnts and have been shown to heterodimerize with Fz in response to non-canonical Wnt binding (*Griffiths et al., 2024*; *Grumolato et al., 2010*). At the same time, like Dvl, Ror2 was reported to bind directly with Vangl2 (*Gao et al., 2011*). We, therefore, postulate that Ror2 may shuttle between Vangl and Fz to deliver Dvl in a Wnt-dependent manner. We note several intriguing links between Ror2 and Dvl2: (1) they both bind to Vangl2 yet display functional antagonism against Vangl2 during CE in over-expression assays (*Figure 1*; *Figure 7—figure supplement 1*); (2) they both cluster with Fz in response to Wnt11 (*Figures 3, 7 and 8*; *Figure 3—figure supplement 3*; *Figure 8—figure supplement 1*), and importantly, Ror2 is required for Dvl2 to dissociate from Vangl2 and cluster with Fz in response to Wnt11 (*Figure 7l*; *Figure 7—figure supplement 2a, a'*). While Ror2 does not seem to bind Dvl directly in our experiment, they both interact with Vangl2, and our SEC data show that Ror2, Vangl2, and Dvl2 co-fractionate (*Figure 7—figure supplement 2*), suggesting that they may form complexes together. Therefore, we envision a model where Vangl acts as an adaptor to bring together Dvl and Ror, either through simultaneous binding of both Dvl and Ror to a single Vangl (*Figure 9a*) or self-oligomerization of Vangl proteins bound separately to Dvl and Ror (*Figure 9b*). When non-canonical Wnt induces Fz-Ror to heterodimerize, the complexes consisting of Ror/Vangl/Dvl can be brought close to Fz to deliver Dvl and initiate non-canonical Wnt signaling (*Figure 9a' and b'*).

In this model, Vangl acts as an unconventional adaptor to simultaneously serve two critical functions: it pre-assembles Ror and Dvl into complexes at the plasma membrane ready to initiate non-canonical Wnt signaling, but at the same time keeps both inactive to prevent ectopic signaling activation. We previously demonstrated that Vangl2 can prevent Dvl from interacting with its downstream effector Daam1 (*Seo et al., 2017*), and our data in this study suggest that Vangl/Pk may act together to sequester both Dvl and Ror from Fz as well (*Figures 4 and 8*; *Figure 4—figure supplement 1*; *Figure 8—figure supplement 1*). Based on this model, Vangl2 overexpression will exert excessive suppression to prevent non-canonical Wnt signaling during CE, which can be overcome by co-overexpressing Dvl2 or Ror2 (*Figure 1*; *Figure 7—figure supplement 1*). Conversely, reducing the dosage of endogenous Vangl2 may decrease the assembly of Ror/Vangl/Dvl complexes, compromising signaling activation in response to non-canonical Wnt (*Figure 8m–o'*). Therefore, our model can explain how partial loss of Vangl2 can synergize with loss of positive non-canonical Wnt signaling regulators, including Ror2, Dvl2, and Wnt5a, to cause various severe CE defects reported in the literature (*Gao et al., 2011*; *Qian et al., 2007*; *Sinha et al., 2012*; *Wang et al., 2011*; *Wang et al., 2006*).

We acknowledge, however, that our model explains primarily the potential molecular actions underlying the regulation of CE at the tissue and genetic levels. Whether and how our model may explain the cellular behavior during CE, such as polarized remodeling of cell junctions or extension of cell protrusions, will require further study.

Lastly, our data implicated an intriguing role for Dvl phosphorylation in the transition from Vangl to Fz. We found that flag-Dvl2 phosphorylation is increased as CE progresses in *Xenopus* and can be elevated by Fz7 but suppressed by Vangl2 (*Figure 6b and c*). In agreement with our over-expression data in *Xenopus*, loss of both Vangl1 and 2 leads to increased Dvl2 and 3 phosphorylation in cell culture (*Mentink et al., 2018*). Intriguingly, Vangl2 seems to bind only to the unphosphorylated form of Dvl2 (*Figure 6d*). These observations suggest that either Dvl phosphorylation per se or another associated modification can be used as a mechanism to decouple Dvl from Vangl. In support of this view, the basic region and PDZ domain of Dsh/Dvl, which mediates Dvl-Vangl interaction (*Angermeier et al., 2025*; *Park and Moon, 2002*), is a strong target of CK1 during PCP signaling in flies (*Klein et al., 2006*; *Strutt et al., 2019*; *Strutt et al., 2006*). Our recent work further revealed that Dvl oligomerization promotes its dissociation from Vangl possibly by occluding the PDZ domain, and Dvl mutants defective at oligomerization also fail to undergo phosphorylation (*Angermeier et al., 2025*).

Taken together, we propose the second piece of our model (*Figure 9c and c'*) that during CE, non-canonical Wnt triggers association of Ror1/2 and Fz to simultaneously accomplish two events: (1) bringing Vangl-sequestered Dvl close to Fz; and (2) activating CK1 or other kinases to phosphorylate Dvl. The combined effects lead to Dvl dissociation from Vangl and transition to Fz in a spatially and temporally controlled manner. On the other hand, by assisting Vangl to sequester Dvl, Pk may suppress the noise from basal CK1 activity and allow cells to respond more specifically and dynamically to non-canonical Wnt signaling during CE.

## Materials and methods

Animal experiments were performed in agreement with the National Institutes of Health. *Xenopus laevis* adults were maintained according to the established protocols by the Institutional Animal Care and Use Committee at the University of Alabama at Birmingham, under Animal Project Number IACUC-22388. There is no evidence for sexual dimorphism in gastrulating *Xenopus* embryos, which were used as the model organism in the study, so sex was not considered as a biological variable in the study design.

### *Xenopus* embryo manipulation and animal cap/DMZ explants

Embryos were acquired by superovulation, maintained in 0.1% MMR solution until the stages for microinjection. Morpholinos or in vitro -synthesized RNAs were injected into either the animal side of two-cell-stage embryos or the DMZ of four-cell-stage embryos. For phenotypic analysis, the DMZ-injected embryos were fixed at tailbud stages, and the dorsal view of embryos was captured using a Leica DFC 490 camera mounted on a Leica M205 FCA stereomicroscope. A Fiji macro was utilized to process the images and obtain the projected area and the length of each embryo. Briefly, the macro first extracts individual embryos from the images, measures the area of each embryo, and generates the smallest rectangle that fully encloses each embryo. The length of the rectangle is considered the length of the embryo, while the width is calculated by dividing the embryo's area by its length. The LWR is then calculated in Excel. The length of the embryos with significant curved shape was corrected manually by drawing a line along the anteroposterior axis to measure the maximal distance from the head to the tail of each embryo using the Leica LAS software with Interactive Measurement module. For the animal cap elongation assay, ectodermal explants were isolated at stages 9–10 and incubated in 0.5 MMR solution containing 10 ng/ml of Activin B (R&D cat# 659-AB-005). The CE phenotype was quantified by measuring the length of the resulting explants. For fluorescent imaging to determine protein localization, DMZ or animal cap explants from injected embryos were isolated at stage 10–10.5, coverslipped and subjected to confocal imaging analysis as described (*Angermeier et al., 2025*; *Seo et al., 2017*).

## Co-immunoprecipitation and western blot

RNAs or morpholinos injected embryo or explants were lysed as described for biochemistry experiments (*Angermeier et al., 2025*; *Seo et al., 2017*; *Tien et al., 2015*). For the co-immunoprecipitation assay, protein lysates were subjected to pull-down with anti-flag (Sigma Anti-FLAG M2 Magnetic Beads (Cat# 8823)) or anti-Myc antibodies (Pierce Cat.#88843), in a buffer containing 50 mM Tris (pH 7.5), 150 mM NaCl, 1 mM EDTA, 10% Glycerol, 0.5% Triton-X100, and 1 x protease inhibitor (Promega #G6521). Western blot detection of proteins was carried out with anti-GFP antibody (Santa Cruz Biotechnology GFP (B-2) (Cat# sc-9996)), anti-myc antibody (Santa Cruz Biotechnology myc Antibody (G-4) (Cat# sc-373712)), anti-Dvl2 (CST Cat.# 3224), anti-XRor2 (Developmental Studies Hybridoma Bank), or anti-flag antibody (Sigma Anti-flag M2 antibody (Cat# F1804)). We followed the protocol by *Burckhardt et al., 2021* and used FIJI to quantify the relative amount of co-IP. Briefly, the intensity values from the co-IP of any protein was divided by that from the IP of its presumptive partner under each control or experimental condition; the mean ratios from the controls was then used to normalize the ratios under each experimental condition to determine the 'Relative Co-IP amount'.

## Fluorescence-detection size-exclusion chromatography (FSEC)

2 cell stage *Xenopus* embryos were co-injected with 3 ng Ror2-EGFP, 0.5 ng Flag-Dvl2, and 1 ng HA-Vangl2 mRNA to the animal side. Animal caps were dissected around stage 10 and cultured in 0.5 X MMR at 15°C overnight. The next day 35–40 animal caps were lysed on ice with the 200 µL lysis buffer (50 mM Tris pH 7.5, 150 mM NaCl, 0.3% Dodecylmaltoside, and Protease inhibitor (Pierce Protease Inhibitor)). After centrifugation at 4°C 14,000 g for 15 min, the supernatant was collected and filtered with a 0.22 mm syringe filter. 100 µL lysate was loaded onto a Superose 6 Increase 10/300 GL column (Cytiva, Marlborough, MA) pre-equilibrated in SEC buffer (Tris 50 mM, pH 7.5, NaCl 150 mM, Dodecylmaltoside 0.03%). The eluted Ror2-GFP was monitored by RF-10AX fluorescence detector (Shimadzu, Japan) following size-exclusion chromatography. Fractions were collected and concentrated (Pierce concentrator PES 10 K MWCO) and used for Western blot.

## Imaging and analyses

For imaging, 0.1–0.5 ng of mRNA encoding Dvl2-mCh, Dvl2-mSc, Dvl2-EGFP, Fz7-EGFP, XRor2-tdTomato, XRor2-EGFP, EGFP-Vangl2, EGFP-Vangl2 RH, flag-XPk, EGFP-mPk2, XWnt11, mem-GFP, and mem-mCh were injected in various combinations into the animal regions at the two-cell stage and dissected at ~St.9, or the DMZ at the four-cell stage and dissected at St. 10.25. Alternatively, *flag-Xpk*-injected animal caps were dissected and fixed in 4% PFA for immunofluorescence staining with an anti-flag antibody. Dissected animal cap or DMZ explants were imaged on an Olympus FV1000 with a 20 x water immersion objective or a Zeiss LSM 900 equipped with Airyscan2 and a 20 x air objective. Images were imported into ImageJ (NIH). Images from at least three different embryos collected on different days were analyzed per injection group.

The relative protein level between the plasma membrane and cytoplasm was quantified by comparing the fluorescent intensity. Briefly, regions corresponding to the plasma membrane and cytoplasm were defined, and the average fluorescence intensity within each region was measured using ImageJ. The ratio of plasma membrane to cytoplasmic fluorescence intensity was then calculated.

Protein enrichment pattern within clusters was analyzed using ImageJ. Briefly, the fluorescence intensity of each pixel along a defined cluster was measured. To account for differences in cluster size, pixels were sequentially numbered from one end of the cluster to the other. The total pixel count for each cluster was normalized to 100, and each pixel was assigned a relative position expressed as a percentile. Similarly, fluorescence intensity was normalized by setting the highest pixel intensity within each cluster to 100 and expressing all other pixel intensities as percentiles of that maximum. A scatter plot was generated with the x-axis representing the relative position and the y-axis representing the normalized fluorescence intensity of each pixel. The regression curve was then fitted to the data to characterize the distribution pattern of fluorescence intensity across the cluster. Ten clusters from 5 to 10 embryos were used for statistical analysis.

To analyze Fz7 endocytosis, the vitelline membrane of injected embryos was removed at stage 10.5, and embryos were incubated in 0.1 X MMR containing 5 µg/ml FM4-64FX (Thermo Fisher, Cat# F34653) for 30 min at room temperature before dissection in 0.1 X MMR. Dissected DMZ explants

were coverslipped, and images were captured without fixation. Images from at least three different embryos collected on different days were analyzed per injection group.

## RNAs and morpholinos

XWnt11, EGFP-Vangl2, HA-Vangl2, GFP-XFz7, flag-XFz7, tdT-tomatoRor2, myc-Vangl2, Dvl2-mCherry, Dvl2-flag, GFP-mPk2, flag-XPk, flag-PL, flag-ΔPL, were transcribed in vitro using mMESSAGE mMACHINE SP6 Transcription Kit (Ambion cat#1340). Xenopus Vangl2-morpholino (XVMO), Ror2-morpholino (Xror2-MO), and Pk-morpholino (XPkMO) are the same as previously described (*Darken et al., 2002*; *Schambony and Wedlich, 2007*; *Takeuchi et al., 2003*). The dosage of each RNA or morpholino is described in each figure.

## Acknowledgements

We thank Dr. Naoto Ueno for providing the flag-XPk construct. The anti-XRor2 monoclonal antibody was obtained from the Developmental Studies Hybridoma Bank, created by the NICHD of the NIH and maintained at The University of Iowa, Department of Biology, Iowa City, IA 52242. This work was supported by grants R35GM131914 (JDA), GM127371 (CC), and HL109130, HL138470, and AR081646 (JW) from the National Institutes of Health.

## Additional information

### Funding

| Funder | Grant reference number | Author |
| --- | --- | --- |
| National Institute of General Medical Sciences | R35GM131914 | Jeffrey D Axelrod |
| National Institute of General Medical Sciences | GM127371 | Chenbei Chang |
| National Heart Lung and Blood Institute | HL109130 | Jianbo Wang |
| National Heart Lung and Blood Institute | HL138470 | Jianbo Wang |
| National Institute of Arthritis and Musculoskeletal and Skin Diseases | AR081646 | Jianbo Wang |

The funders had no role in study design, data collection and interpretation, or the decision to submit the work for publication.

### Author contributions

Hwa-seon Seo, Conceptualization, Data curation, Formal analysis, Investigation, Visualization, Writing – original draft; Deli Yu, Data curation, Formal analysis, Validation, Investigation, Visualization, Methodology, Writing – review and editing; Ivan K Popov, Formal analysis, Investigation, Methodology; Jiahui Tao, Formal analysis, Investigation; Allyson R Angermeier, Fei Yang, Investigation, Methodology; Sylvie Marchetto, Jean-Paul Borg, Resources; Bingdong Sha, Conceptualization, Formal analysis; Jeffrey D Axelrod, Conceptualization, Resources, Writing – review and editing; Chenbei Chang, Conceptualization, Resources, Funding acquisition, Investigation, Writing – review and editing; Jianbo Wang, Conceptualization, Resources, Data curation, Formal analysis, Supervision, Funding acquisition, Investigation, Visualization, Methodology, Writing – original draft, Project administration, Writing – review and editing

### Author ORCIDs

Deli Yu ⬥ https://orcid.org/0000-0002-9673-0870
Jiahui Tao ⬥ https://orcid.org/0000-0002-3016-1249
Jean-Paul Borg ⬥ https://orcid.org/0000-0001-8418-3382

Jeffrey D Axelrod ⓘ https://orcid.org/0000-0001-6094-7392
Jianbo Wang ⓘ https://orcid.org/0000-0002-6769-9851

### Ethics

Animal experiments were performed in agreement with the National Institutes of Health. Xenopus laevis adults were maintained according to the established protocols by the Institutional Animal Care and Use Committee at the University of Alabama at Birmingham, under Animal Project Number IACUC-22388.

Reviewer #1 (Public review): https://doi.org/10.7554/eLife.91199.4.sa1
Author response https://doi.org/10.7554/eLife.91199.4.sa2

## Additional files

### Supplementary files
MDAR checklist

### Data availability
All relevant data and resources can be found within the article and its supplementary information.

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
