## [Editor Report · eLife Assessment]

This **valuable** study addresses mechanisms of feedback inhibition between planar cell polarity protein complexes during convergent extension movements in Xenopus embryos. The authors propose a conceptually new model, in which non-canonical Wnt ligand stimulates the transition of Dishevelled from its complex with Vangl to Frizzled, with essential roles of Prickle and Ror in this process. The main observations supporting molecular interactions rely on modest but significant changes in protein association in response to Wnt11. While the study is limited due to insufficient phenotypic analysis at the cellular level and the use of exogenously supplied proteins, this work is **convincing** and will be of broad interest to cell and developmental biologists.

---

## [Referee Report · Reviewer #1 (Public review)]

Summary:

Planar cell polarity core proteins Frizzled (Fz)/Dishevelled (Dvl) and Van Gogh-like (Vangl)/Prickle (Pk) are localized on opposite sides of the cell and engage in reciprocal repression to modulate cellular polarity within the plane of static epithelium. In this interesting manuscript, the authors explore how the anterior core proteins (Vangl/Pk) inhibit the posterior core protein (Dvl). The authors propose that Pk assists Vangl2 in sequestering both Dvl2 and Ror2, while Ror2 is essential for Dvl to transition from Vangl to Fz in response to non-canonical Wnt signaling.

Strengths:

The strengths of the manuscript are found in the very interesting and new concept along with supportive data for a model of how non-canonical Wnt induces Dvl to transition from Vangl to Fz with an opposing role for PK and Vangl2 to suppress Dvl during convergent extension movements. Ror is key player required for the transition and antagonizes Vangl.

Weaknesses:

In addition to general whole embryo morphology that is used as evidence for CE defects, two forms of data are presented: co-expression and IP, as well as IF of exogenously expressed proteins. The microscopy would benefit from super-resolution microscopy since in many cases the differences in protein localization are not very pronounced, and Western analysis data often show relatively subtle differences. Thus, future work will determine the strength of the interactions of the model.

Major points.

Overexpression conditions

A possible concern is that most analyses were performed with overexpression conditions. PCP core proteins (Vangl2, Pk, Dvl, and Fz receptors) are known to display polarized subcellular localization in both the neural epithelium and DMZ explants (Ref: PCP and Septins govern the polarized organization of the actin cytoskeleton during convergent extension, Current Biology, 2024). However, in this study, overexpressed PCP core proteins failed to show polarized localization. Thus, one must be careful in interpreting data.

Subtle effects

Several of the reported results show quite modest changes in imaging and immunoprecipitation analyses, which are supportive of the proposed molecular model, but future experiments will be needed to robustly test the model.

---

## [Author Response]

The following is the authors’ response to the previous reviews

**Public Review:**

**Reviewer #1 (Public review):**
The weaknesses are in the clarity and resolution of the data that forms the basis of the model. In addition to general whole embryo morphology that is used as evidence for CE defects, two forms of data are presented, co-expression and IP, as well as a strong reliance on IF of exogenously expressed proteins. Thus, it is critical that both forms of evidence be very strong and clear, and this is where there are deficiencies; (1) For vast majority of experiments general morphology and LWR was used as evidence of effects on convergent extension movements rather than keller explants or actual cell movements in the embryo. (2) the microscopy would benefit from super resolution microscopy since in many cases the differences in protein localization are not very pronounced. (3) the IP and Western analysis data often shows very subtle differences, and some cases not apparent.Major points.(1) Assessment of CE movementThe authors conducted an analysis of the subcellular localization of PCP core proteins, including Vangl2, Pk, Fz, and Dvl, within animal cap explants (ectodermal explants). The authors primarily used the length-to-width ratio (LWR) to evaluate CE movement as a basis for their model. However, LWR can be influenced by multiple factors and is not sufficient to directly and clearly represent CE defects. While the author showed that Prickle knockdown suppresses animal cap elongation mediated by Activin treatment, they did not test their model using standard assays such as animal cap elongation or dorsal marginal zone (DMZ) Keller explants. Furthermore, although various imaging analyses were performed in Wnt11-overexpressing animal caps and DMZ explants, the Wnt11-overexpressing animal caps did not undergo CE movement. Given that this study focuses on the molecular mechanisms of Vangl2 and Ror2 regulation of Dvl2 during CE, the model should be validated in more appropriate tissues, such as DMZ explants.(2) Overexpression conditionsAnother concern is that most analyses were performed with overexpression conditions. PCP core proteins (Vangl2, Pk, Dvl, and Fz receptors) are known to display polarized subcellular localization in both the neural epithelium and DMZ explants (Ref: PCP and Septins govern the polarized organization of the actin cytoskeleton during convergent extension, Current Biology, 2024). However, in this study, overexpressed PCP core proteins failed to show polarized localization. Previous studies, such as those from the Wallingford lab, typically used 10-30 pg of RNA for PCP core proteins, whereas this study injected 100-500 pg, which is likely excessive and may have created artificial conditions that confound the imaging results.(3) Subtle and insufficient effectsSeveral of the reported results show quite modest changes in imaging and immunoprecipitation analyses, which are not sufficient to strongly support the proposed molecular model. For example, most Dvl2 remained localized with Fz7 even under Vangl2 and Pk overexpression (Fig. 4). Similarly, Wnt11 overexpression only slightly reduced the association between Vangl2 and Dvl2 (Sup. Fig. 8), and the Ror2-related experiments also produced only subtle effects (Fig. 8, Sup. Fig. 15).

We thank reviewer 1 for careful reading of our revised manuscript, and additional constructive criticisms. Since the two reviewers had divergent opinions towards our revised manuscript, we think that it might be more productive to request a Version of Record at this point, and have our proposed model debated/ tested by others in the field. We will keep the reviewer’s suggestions in mind while design ongoing studies. We would like to address the criticisms collectively below:

(1) The primary goal of our current manuscript is to build a mechanistic model for non-canonical Wnt signaling through elucidating the functional relationships between Dvl, Vangl, PK and Ror during CE. They each have been studied extensively in prior literature using DMZ injected embryos, and DMZ, Keller and animal cap explants, so there is little doubt that the reduced LWR following their over-expression or knockdown in DMZ is due to disruption of CE. In the context of our study in the current manuscript, we primarily performed their co-injections in different combinations to differentiate synergistic vs. antagonistic relationship, and in the majority cases we relied on epistatsis to draw conclusions (e.g. Fig. 1; Fig. 2h, I; Suppl. Fig. 6; Suppl. Fig. 14). Nevertheless, we did follow the reviewer’s suggestion and used animal cap elongation as an additional assay to confirm that Pk and Vangl2 did synergize to disrupt CE, and their synergy could be blocked by Dvl2 co-overexpression; the new data is added to Fig. 1 (Fig. 1h, h’). Therefore, given the prior literature, our new animal cap explant data, and the specific scope of our current study, we feel that the LWR measurement is a reasonable assay to determine CE phenotype in this manuscript. We fully agree with the reviewer that our model will need to be tested at the cellular level through live imaging of DMZ explants; it is indeed the direction of our future study, but is beyond the scope of the current manuscript.

(2) A salient feature of non-canonical Wnt signaling is that loss or over-expression of any components can often cause identical CE defects at the tissue/ embryo level. We used many co-injection experiments to demonstrate that this is due, at least in part, to a counterbalance between Dvl/Ror and Vangl/PK (e.g. Fig. 1; Fig. 2h, I; Suppl. Fig. 6; Suppl. Fig. 14). It is in this context that we planned the imaging and biochemical experiments to determine the possible molecular mechanisms underlying their functional interaction, and we feel that the moderate over-expression used is reasonable in this case for us to build the first integrated model. We do plan to test our model using lower expression in the future. To acknowledge the limitation of our study, we also added the following sentences in the Discussion:

“We acknowledge, however, that our model explains primarily the potential molecular actions underlying the regulation of CE at the tissue level. Whether and how our model may explain the cellular behavior during CE, such as polarized remodeling of cell junction or extension of cell protrusions, will require further study.”

(3) The Wnt11 induced reduction of Dvl2-Vangl2 co-IP (Suppl. Fig. 8, 15) may be moderate, but is statistically significant and reproducible, and we have reported similar findings in two other publications (DOI: 10.1093/hmg/ddx095; DOI: 10.1038/s41467-025-57658-0). Given the limitation of co-IP, we had to rely on high level over-expression to make the experiments feasible. We are building proximity based assays such as NanoBRET, and plan to verify the result with lower level expression in the future.

**Reviewer #2 (Public review):**

We thank the reviewer for the encouraging comments, and the suggestion to clarify the description related to Suppl. Fig. 15. We made revision according to the reviewer’s suggestion, and added Suppl. Fig. 16 to further examine the effect of Ror2 knockdown on the steady state interaction between Dvl2 and Vangl2 using imaging approach.